# Learning Hierarchical World Models with Adaptive Temporal Abstractions from Discrete Latent Dynamics

**Christian Gumbsch**[1,2*]**, Noor Sajid**[3]**, Georg Martius**[2] **& Martin V. Butz**[1]

[1] Neuro-Cognitive Modeling, University of Tübingen, Tübingen, Germany

[2] Autonomous Learning, Max Planck Institute for Intelligent Systems, Tübingen, Germany

[3] Wellcome Centre for Human Neuroimaging, University College London, London, U.K.

[*] corresponding author: `christian.gumbsch@uni-tuebingen.de`

## Abstract

Hierarchical world models can significantly improve model-based reinforcement learning (MBRL) and planning by enabling reasoning across multiple time scales. Nonetheless, the majority of state-of-the-art MBRL methods employ flat, non-hierarchical models. We propose Temporal Hierarchies from Invariant Context Kernels (THICK), an algorithm that learns a world model hierarchy via discrete latent dynamics. The lower level of THICK updates parts of its latent state sparsely in time, forming invariant contexts. The higher level exclusively predicts situations involving context changes. Our experiments demonstrate that THICK learns categorical, interpretable, temporal abstractions on the high level, while maintaining precise low-level predictions. Furthermore, we show that the emergent hierarchical predictive model seamlessly enhances the abilities of MBRL or planning methods. We believe that THICK contributes to the further development of hierarchical agents capable of more sophisticated planning and reasoning abilities.

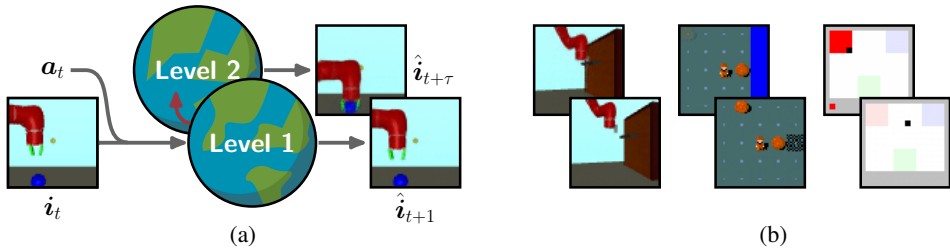

Figure 1: **THICK world models** predict on two levels. (a) Level 1 predicts the next input $(t + 1)$. Level 2 predicts a future state $(t + \tau)$ expected to change an otherwise constant latent state. (b) Exemplary low- (bottom) and high-level predictions (top) for: opening a door (`Multiworld-Door`), pushing a boulder into water (`Minihack-River`), or activating a pad (`VisualPinPadThree`).

## 1 Introduction

The intricate hierarchical representations formed in our brains through sensorimotor experience (Lee & Mumford, 2003; Rougier et al., 2005; Botvinick & Weinstein, 2014; Rohe & Noppeney, 2015; Lake et al., 2017; Friston et al., 2018; 2021; Radvansky & Zacks, 2014; Butz, 2016; Tomov et al., 2020) serve as a useful blueprint for enhancing the planning abilities of artificial agents through hierarchical world models (Schmidhuber, 1992; LeCun, 2022). Humans, for example, can plan their behavior on various time scales and flexibly switch between them, such as picking up a pen to write an invitation when organizing a party.

Despite recent advances in equipping MBRL agents with the capacity to learn world models, i.e., autonomously learned forward models that encode the interaction of an agent with its environment (Ha & Schmidhuber, 2018; Hafner et al., 2019b;a; 2020; 2023), these models lack a hierarchical structure. Consequently, they are restricted to predictions on predefined time scales, hampering their capability for long-horizon planning. The main challenge lies in formalizing suitable methods for

learning higher-level abstractions (Sutton, 1988; Sutton et al., 1999; Eppe et al., 2022; Precup, 2000; van Seijen et al., 2014). Importantly, these abstractions should be tied neither to particular tasks nor to fixed nested time scales. Context-conditioned, event-predictive structures offer themselves as temporally flexible, basic compositional unit (Butz, 2016; Heald et al., 2021; 2023).

We present a deep learning architecture that learns hierarchical world models, which we call **Temporal Hierarchies from Invariant Context Kernels** (THICK[1]). THICK adaptively discovers higher-level time scales by guiding the lower-level world model to update parts of its latent state only sparsely in time. The high-level model is then trained to predict scenarios involving changes in these low-level latent states. A depiction of THICK world models can be found in Fig. 1a.

We make the following key contributions:

- We introduce the Context-specific Recurrent State Space Model (C-RSSM), which enhances Dreamer's (Hafner et al., 2019b; 2020) Recurrent State Space Model (RSSM) by **encoding context-sensitive dynamics** via sparsely changing latent factors, labeled *context*.
- We introduce THICK, which learns a **hierarchy of world models**. The high-level runs at an adaptive time scale developing **higher-level actions** that anticipate lower-level context changes.
- We demonstrate the effectiveness of THICK in two **planning** scenarios: $i$) using THICK's hierarchical predictions to enhance **MBRL in long-horizon tasks**, and $ii$) using THICK's high-level predictions to set subgoals for **hierarchical model-predictive planning** (MPC).

## 2 METHOD

### 2.1 C-RSSM WORLD MODEL

The RSSM proposed in Hafner et al. (2019b) is a recurrent neural network (RNN) that is used for model-based reinforcement learning (Hafner et al., 2019a; 2020; 2023; Sekar et al., 2020; Mendonca et al., 2021; Sajid et al., 2021). RSSM embeds input images $i_t$ and actions $a_t$ into a latent state $s_t$ and predicts dynamics exclusively within this state. All aspects of the latent state evolve continuously. We require sparse latent state changes to establish hierarchical world models. Accordingly, our **Context-specific RSSM** (C-RSSM) integrates a sparsely changing latent state $c_t$ as context with a coarse prediction pathway (cf. Fig. 2). Our C-RSSM with trainable parameters $\phi$ is computed by:

$$\text{Latent state:} \quad s_t \leftarrow [c_t, h_t, z_t] \quad (1) \qquad \text{Pre. Prior:} \quad \hat{z}_t^h \sim p_\phi^h\big(\hat{z}_t^h \mid c_t, h_t\big) \quad (4)$$

$$\text{Coarse Dyn.:} \quad c_t = g_\phi(a_{t-1}, c_{t-1}, z_{t-1}) \quad (2) \qquad \text{Coa. Prior:} \quad \hat{z}_t^c \sim p_\phi^c\big(\hat{z}_t^c \mid a_{t-1}, c_t, z_{t-1}\big) \quad (5)$$

$$\text{Pre. Dyn.:} \quad h_t = f_\phi(a_{t-1}, c_t, h_{t-1}, z_{t-1}) \quad (3) \qquad \text{Posterior:} \quad z_t \sim q_\phi(z_t \mid c_t, h_t, i_t) \quad (6)$$

Equations in red are exclusive to C-RSSM.[2] We separate RSSM's latent state $s_t$ into three parts (Eq. 1): a stochastic state $z_t$, a continuously updated, high-dimensional, deterministic state $h_t$, and a sparsely changing, low-dimensional context $c_t$. At time $t$ the C-RSSM first updates the context $c_t$ (Eq. 2), where actual $c_t$ changes only occur *sparsely in time*. Next, C-RSSM updates $h_t$ via a GRU (Chung et al., 2014) cell $f_\phi$ (Eq. 3). The C-RSSM makes two predictions about the next stochastic state $\hat{z}_t^h$: $i$) a *precise* prior based on both $h_t$ and $c_t$ (Eq. 4), and $ii$) a *coarse* prior $\hat{z}_t^c$ based on the context, stochastic state, and action, ignoring $h_t$ (Eq. 5). Given the input image $i_t$, C-RSSM updates its posterior $z_t$ (Eq. 6). Following DreamerV2 (Hafner et al., 2020), we sample $z_t$ from a vector of categorical distributions. Note that Eq. 2 and Eq. 5 do not depend on $h_{t-1}$, thus creating a **coarse processing pathway** independent of $h$. This enables predictions using only $c_t$ as a deterministic memory, which is crucial because it $i$) encourages encoding prediction-relevant information in $c_t$ and $ii$) allows predictions without $h_t$ which we will use later (details in Suppl. D.1).

Besides encoding latent dynamics, C-RSSM is trained to reconstruct observable variables $y_t$ of the outside world from its latent states $s_t$. Two output heads $o_\phi$ generate precise and coarse predictions:

$$\text{Precise prediction:} \quad \hat{y}_t \sim o_\phi(\hat{y}_t \mid s_t) \quad (7) \qquad \text{Coarse prediction:} \quad \hat{y}_t^c \sim o_\phi^c(\hat{y}_t^c \mid c_t, z_t). \quad (8)$$

We predict the input image $i_t$, the reward $r_t$, and reward discount $\gamma_t$[3], i.e., $y_t \in \{i_t, r_t, \gamma_t\}$.

---

[1]In philosophy, the term 'thickness' refers to concepts that combine descriptions with an evaluative context (Roberts, 2013). A THICK world model fuses representations of the world with a contextual interpretation.

[2]Removing $c$ in all black equations recovers the equations for the RSSM (Eqns. 1,3,4,6).

[3]The discount $\gamma_t$ is set to 0 if an episode terminates and to a fixed value $\gamma$ otherwise.

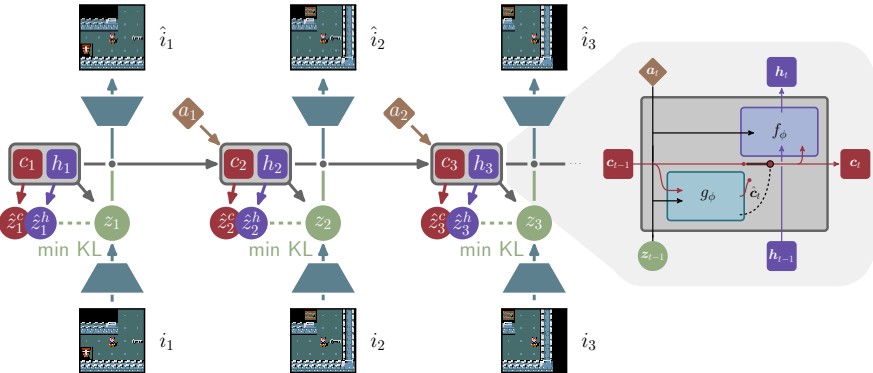

Figure 2: **C-RSSM world model**. Left: The C-RSSM encodes dynamics within latent states with a stochastic part $z_t$ and two deterministic parts $h_t$ and $c_t$ The network predicts the next stochastic state $z_t$ via two pathways: It makes *coarse* predictions $\hat{z}_t^c$ based mainly on $c_t$ and *precise* predictions $\hat{z}_t^h$ based on $h_t$. Right: Internally, the sparsely changing context $c_t$ is updated via a GateL0RD cell $g_\phi$ with a sparsely operated update gate. A GRU cell $f_\phi$ is used to continuously change $h_t$.

**Sparse context updates**    The latent context code $c_t$ is designed to change sparsely in time, ideally at distinct, environment-specific, transition points. Accordingly, the coarse dynamics $g_\phi$ (Eq. 2) are modeled by a GateL0RD cell (Gumbsch et al., 2021), which learns sparsely changing latent states $c_t$ via an update gate, whose activity is $L_0$-regularized via loss term $\mathcal{L}^{\mathrm{sparse}}$. Note that context $c_t$ alone is too coarse to predict the current stochastic state $z_t$.

**Loss function**    Given a sequence of length $T$ of input images $i_{1:T}$, actions $a_{1:t}$, rewards $r_{1:T}$, with discounts $\gamma_{1:T}$, the parameters $\phi$ of C-RSSM are jointly optimized to minimize the loss $\mathcal{L}(\phi)$:

$$\mathcal{L}(\phi) = \mathbb{E}_{q_\phi}\big[\beta^{\mathrm{pred}}\mathcal{L}^{\mathrm{pred}}(\phi) + \beta^{\mathrm{KL}}\mathcal{L}^{\mathrm{KL}}(\phi) + \beta^{\mathrm{sparse}}\mathcal{L}^{\mathrm{sparse}}(\phi)\big], \qquad (9)$$

including the prediction loss $\mathcal{L}^{\mathrm{pred}}$, the KL loss $\mathcal{L}^{\mathrm{KL}}$, and sparsity loss $\mathcal{L}^{\mathrm{sparse}}$ with respective hyperparameters $\beta^{\mathrm{pred}}$, $\beta^{\mathrm{KL}}$, and $\beta^{\mathrm{sparse}}$. The prediction loss $\mathcal{L}^{\mathrm{pred}}$ drives the system to accurately predict perceptions $y$ via its output heads $o_\phi$, including context-conditioned coarse predictions (Eq. 8). The KL loss $\mathcal{L}^{\mathrm{KL}}$ minimizes the KL divergences between prior predictions $p_\phi^h$ and $p_\phi^c$ and the approximate posterior $q_\phi$. The sparsity loss $\mathcal{L}^{\mathrm{sparse}}$ encourages consistency of context $c_t$. The exact loss functions are provided in Suppl. D.3. We set $\beta^{\mathrm{pred}}$ and $\beta^{\mathrm{KL}}$ to DreamerV2 defaults (Hafner et al., 2020) and modify the sparsity loss scale $\beta^{\mathrm{sparse}}$ depending on the scenario (cf. Suppl. B).

## 2.2    HIERARCHICAL WORLD MODEL

To learn a hierarchical world model, we leverage C-RSSM's discrete context $c_t$ updates by means of our **Temporal Hierarchies from Invariant Context Kernels** (THICK) algorithm. A C-RSSM world model $w_\phi$ segments sequences into periods of stable context activity ($c_t = c_{t+1} = \cdots = c_{\tau-1}$), interspersed with sparse context updates (cf. Fig. 3a). THICK uses these discrete context dynamics as an **adaptive timescale** for training a high-level network $W_\theta$. The core assumption is that states prompting context updates coincide with crucial changes in latent generative factors. These key states are predicted by the high-level network $W_\theta$, while states between context updates are ignored.

To train the high-level world model $W_\theta$, we require input-target pairs for a given sequence of $T$ images $i_{1:T}$, actions $a_{1:T}$, and episode termination flags $d_{1:T}$. The sequence is passed through the low-level model $w_\phi$ to obtain a sequence of contexts $c_{1:T}$. Targets are defined as all time steps $\tau$ with context changes, i.e., where $c_\tau \neq c_{\tau-1}$ or the episode ends. We define the function $\tau(\cdot)$ as

$$\tau(t) = \min\big(\{\tau \mid \tau > t \wedge (c_\tau \neq c_{\tau-1} \vee d_\tau = 1)\}\big). \qquad (10)$$

Thus, $\tau(\cdot)$ maps every point $t$ to the next point in time $\tau(t)$ with context change, effectively implementing a variable temporal abstraction that generates target predictions $\tau(t)$ for every $t$.

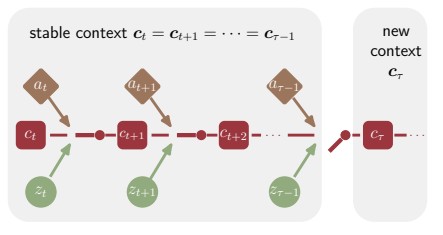 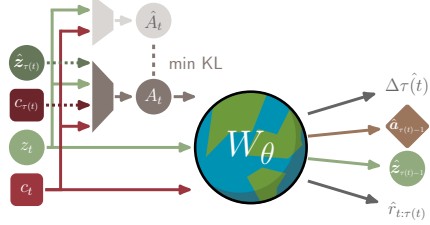

(a) Time series discretization                    (b) Level 2 world model

Figure 3: **High-level segmentation**: (a) The low-level C-RSSM discretizes sequences into segments with constant contexts. We use this segmentation to determine inputs and targets for the high level. (b) The high-level world model predicts the states and actions that lead to a context change at time $\tau(t)$ from latent states $\mathbf{z}_t$ and $\mathbf{c}_t$. High-level actions ($\mathbf{A}_t$ or $\hat{\mathbf{A}}_t$) distinguish high-level outcomes.

**High-level targets**   We predict all variables at $\tau(t)$ that may cause a context change or are needed for planning across a context change: $\hat{\mathbf{z}}_{\tau(t)-1}, \hat{\mathbf{a}}_{\tau(t)-1}, \Delta\hat{\tau}(t), \hat{r}^\gamma_{t:\tau(t)}$ (cf. Fig. 3b). In particular, we predict the stochastic states $\hat{\mathbf{z}}_{\tau(t)-1}$ and actions $\hat{\mathbf{a}}_{\tau(t)-1}$ immediately before a context change at time $\tau(t)$, because both can cause an update of $\mathbf{c}_{\tau(t)}$ (see Eq. 2). Intuitively, this means that observations, e.g. seeing something fall, as well as actions, e.g. catching something, could contribute to a change of $\mathbf{c}_t$. We furthermore predict the elapsed time $\Delta\tau(t)$ and the accumulated discounted reward $r^\gamma_{t:\tau(t)}$, which may account for variable duration and rewards when evaluating high-level outcomes:

$$\text{Elapsed time:} \quad \Delta\tau(t) = \tau(t) - t \qquad \text{Accumulated rewards:} \quad r^\gamma_{t:\tau(t)} = \sum_{\delta=0}^{\Delta\tau(t)-1} \gamma^\delta r_{t+\delta} \quad (11)$$

**High-level inputs**   To predict high-level targets, we use the low-level stochastic state $\mathbf{z}_t$ and context $\mathbf{c}_t$ as inputs. However, we need to disambiguate different potential outcomes, which generally depend on the world and the policy pursued by the agent. Accordingly, akin to actions on the low level, we create self-organizing high-level "actions" $\mathbf{A}_t$, similar to skills or options (Sutton et al., 1999). $\mathbf{A}_t$ encode a categorical distribution over probable next context changes. To learn $\mathbf{A}_t$, the high-level world model implements a *posterior* action encoder $Q_\theta$ and a *prior* action encoder $P_\theta$ (cf. Fig. 3b). Overall, the high-level world model $W_\theta$ with learnable parameters $\theta$ is computed by:

$$\text{Post.:} \ \mathbf{A}_t \sim Q_\theta(\mathbf{A}_t \mid \mathbf{c}_t, \mathbf{z}_t, \mathbf{c}_{\tau(t)}, \mathbf{z}_{\tau(t)}) \quad (12) \qquad \text{Prior:} \ \hat{\mathbf{A}}_t \sim P_\theta(\hat{\mathbf{A}}_t \mid \mathbf{c}_t, \mathbf{z}_t) \quad (15)$$

$$\text{Action:} \ \hat{\mathbf{a}}_{\tau(t)-1} \sim F_\theta^{\hat{\mathbf{a}}}(\hat{\mathbf{a}}_{\tau(t)-1} \mid \mathbf{A}_t, \mathbf{c}_t, \mathbf{z}_t) \quad (13) \qquad \text{Time :} \ \Delta\hat{\tau}(t) \sim F_\theta^{\hat{\tau}}(\Delta\hat{\tau}(t) \mid \mathbf{A}_t, \mathbf{c}_t, \mathbf{z}_t) \quad (16)$$

$$\text{State:} \ \hat{\mathbf{z}}_{\tau(t)-1} \sim F_\theta^{\hat{\mathbf{z}}}(\hat{\mathbf{z}}_{\tau(t)-1} \mid \mathbf{A}_t, \mathbf{c}_t, \mathbf{z}_t) \quad (14) \qquad \text{Reward:} \ \hat{r}^\gamma_{t:\tau(t)} \sim F_\theta^{\hat{r}}(\hat{r}^\gamma_{t:\tau(t)} \mid \mathbf{A}_t, \mathbf{c}_t, \mathbf{z}_t) \quad (17)$$

The posterior $Q_\theta$ receives not only $\mathbf{c}_t$ and $\mathbf{z}_t$ as its input but also privileged information about the actually encountered next context, i.e. $\mathbf{c}_{\tau(t)}$ and $\mathbf{z}_{\tau(t)}$ (Eq. 12), which leads to the emergence of individualized, result-conditioned action encodings in $\mathbf{A}_t$. The prior $P_\theta$ learns a distribution over $\hat{\mathbf{A}}_t$ approximating the posterior without the privileged information (Eq. 15). During training, THICK samples the high-level action $\mathbf{A}_t$ from $Q_\theta$. During evaluation, we sample from the prior $P_\theta$ instead. We model $\hat{\mathbf{A}}_t$ and $\mathbf{A}_t$ as one-hot encoded categorical variables.

**Loss function**   The high-level world model $W_\theta$ with parameters $\theta$ is trained to minimize the loss

$$\mathfrak{L}(\theta) = \mathbb{E}\left[\alpha^{\text{pred}}\mathfrak{L}^{\text{pred}}(\theta) + \alpha^{\mathbf{A}}\mathfrak{L}^{\mathbf{A}}(\theta)\right], \quad (18)$$

with hyperparameters $\alpha^{\text{pred}}$ and $\alpha^{\mathbf{A}}$ scaling the prediction $\mathfrak{L}^{\text{pred}}$ and action $\mathfrak{L}^{\mathbf{A}}$ loss terms, respectively. The prediction loss drives the system to better predict the high-level targets. The action loss drives the system to minimize the KL divergence between the posterior high-level action distribution $Q_\theta$ and the prior distribution $P_\theta$. The exact loss functions can be found in Suppl. D.4.

**Summary**   Our THICK world model augments traditional flat world models by a high level, which learns predictions of variable length, anticipating context transitions. This augmentation allows for seamless transitions between coarse, low-level and abstract, high-level predictions. Given a

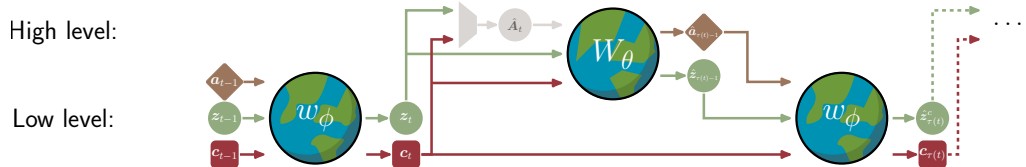

High level:

Low level:

**Figure 4: Temporal abstract predictions of THICK world models.** From a low-level context $c_t$ and stochastic state $z_t$, the high level predicts a future stochastic state $\hat{z}_{\tau(t)-1}$ as well as the action $\hat{a}_{\tau(t)-1}$. With these predictions, the context $c_{\tau(t)}$ is updated on the low level together with the coarse prior $\hat{z}^c_{\tau(t)}$. This process can be repeated (dashed line) to create a temporal abstract roll-out.

context $c_t$, stochastic state $z_t$, and sampled high-level action $\hat{A}_t$, the high-level model $W_\theta$ predicts a scenario $(\hat{a}_{\tau(t)-1}, \hat{z}_{\tau(t)-1})$ immediately prior to the next anticipated context change. By feeding this prediction into the coarse processing pathway of C-RSSM, we can predict the subsequent, new context $c_{\tau(t)}$ (Eq. 2) and a coarse prior estimate of the corresponding stochastic state $\hat{z}^c_{\tau(t)}$ (Eq. 5). Longer *temporal abstract roll-outs* can be created by feeding $c_{\tau(t)}$ and $\hat{z}^c_{\tau(t)}$ again into $W_\theta$ (see Fig. 4). In this way, actual context change predictions are naturally generated by C-RSSM.

## 2.3 DOWNSTREAM APPLICATIONS OF THICK WORLD MODELS

World models have been applied in many downstream tasks, including MBRL (Ha & Schmidhuber, 2018; Hafner et al., 2019a; 2020; 2023), exploration (Sekar et al., 2020; Sancaktar et al., 2022), or model-predictive control (MPC) (Hafner et al., 2019b; Vlastelica et al., 2021). With minimal changes, the hierarchical roll-outs from THICK can be seamlessly integrated where flat roll-outs were previously utilized. We exemplify this integration in two key areas: MBRL and MPC.

### 2.3.1 THICK DREAMER: MBRL WITH HIERARCHICAL ROLLOUTS

Dreamer (Hafner et al., 2019a) learns behavior by training an actor and a critic from "imagined" roll-outs of its RSSM world model. More specifically, Dreamer imagines a sequence of states $s_{t:t+H}$ from a start state $s_t$ given an actor-generated action sequence $a_{t:t+H}$. Dreamer computes the general $\lambda-$return $V^\lambda(s_t)$ (Sutton & Barto, 2018) for every $s_t$ and its critic $v_\xi$ is trained to regress $V^\lambda(s_t)$.

In sparse reward tasks, one challenge is reward propagation for training the critic (Andrychowicz et al., 2017). Here, Dreamer faces a difficult trade-off: Long roll-outs (large $H$) speed up reward propagation but degrade the quality of the predicted roll-outs. We propose **THICK Dreamer**, which combines value estimates from low- and high-level predictions to boost reward propagation. THICK Dreamer maintains an additional critic $v_\chi$ to evaluate temporal abstract predictions. Like Dreamer, we first imagine a low-level roll-out of $H$ states $s_{t:t+H}$. Additionally, for every time $t$ in the roll-out, we predict a temporal abstract outcome $c_{\tau(t)}$ and $z_{\tau(t)}$ and estimate a long horizon value $V^{\text{long}}$ as

$$V^{\text{long}}(s_t) = \hat{r}^\gamma_{t:\tau(t)} + \gamma^{\Delta\hat{t}}\left(\hat{r}^c_{\tau(t)} + \hat{\gamma}^c_{\tau(t)}v_\chi(\hat{c}_{\tau(t)}, \hat{z}_{\tau(t)})\right), \tag{19}$$

with all variables predicted via the THICK world model and immediate rewards via Eq. 8 of C-RSSM given THICK's world model predictions (cf. also supplementary Alg. 1). We estimate the value of a state $s_t$ as a mixture of short- and long-horizon estimates with

$$V(s_t) = \psi V^\lambda(s_t) + (1-\psi)V^{\text{long}}(s_t), \tag{20}$$

where the hyperparameter $\psi$ controls the trade-off between the two estimates. We set $\psi = 0.9$ in all experiments and train both critics $v_\xi$ and $v_\chi$ to regress the value estimate. In sum, to speed up credit assignment when learning a value function, THICK Dreamer combines low-level roll-outs with temporal abstract predictions to additionally estimate the value of likely long-horizon outcomes.

### 2.3.2 THICK PLANET: HIERARCHICAL MPC

The original RSSM was proposed in PlaNet (Hafner et al., 2019b) as a world model for MPC. PlaNet searches for the optimal action sequence $a^*_{t:t+H}$ to maximize the predicted returns $\hat{r}_{t:t+H}$. Thereby, PlaNet employs zero-order trajectory optimization via the cross entropy method (CEM) (Rubinstein, 1999). Once $a^*_{t:t+H}$ is identified, the initial action $a^*_t$ is executed and the procedure is repeated.

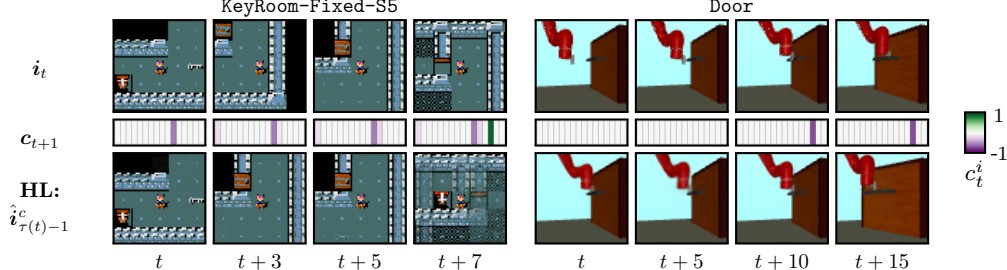

Figure 5: **Context changes**. We show the input images $i_t$, 16-dim. contexts $c_{t+1}$ and reconstructed high-level predictions $\hat{i}^c_{\tau(t)-1}$. For KeyRoom the context changes when finding the key, picking it up, opening a door (here from a diagonally adjacent grid) or exiting the room. In Door the context changes when the robot grabs the handle. The high level predicts the states before the next changes.

CEM optimizes randomly sampled trajectories. Sampling a good action sequence is exponentially harder for increasing task horizons. We hypothesize that such tasks could be solved with much fewer high-level actions. For this, we propose **THICK PlaNet**. THICK PlaNet plans on the high level to solve the task and uses the low level to follow this plan. We define a reward function $R(\cdot)$ to estimate the return of a high-level action sequence $\boldsymbol{A}_{1:K}$ with length $K$ recursively as

$$R(\boldsymbol{A}_{k:K}, t') = \hat{r}^\gamma_{t':\tau(t')} + \gamma^{\Delta \hat{t}} \begin{cases} \hat{r}^c_{\tau(t')} + \hat{\gamma}^c_{\tau(t')} R(\boldsymbol{A}_{k+1:K}, \tau(t')+1) & \text{for } k < K, \\ \hat{r}^c_{\tau(t')} & \text{for } k = K \end{cases} \quad (21)$$

with all variables predicted via a temporal abstract roll-out (see Sec. 2.2) starting with $k = 1$ and $t' = t$. We search for the optimal sequence $\hat{\boldsymbol{A}}^*_{1:K}$ maximizing $R(\cdot)$ with Monte Carlo Tree Search. Based on the first action $\hat{\boldsymbol{A}}^*_1$ we sample a subgoal $\hat{z}^{\text{goal}}_t \sim F_\theta(\hat{z}^{\text{goal}}_t | \hat{\boldsymbol{A}}^*_1, c_t, z_t)$. This subgoal is valid as long as it has not been reached yet and nothing has drastically changed in the environment. Thus, we only replan on the high level when the context has changed. We apply CEM on the low level to reach $z^{\text{goal}}_t$ while also maximizing task return with

$$\boldsymbol{a}^*_{t:t+H} = \arg\max_{\boldsymbol{a}_{t:t+H}} \sum_{t'=t}^{t+H} \hat{r}_{t'} + \kappa \sin(\boldsymbol{z}_{t'}, \boldsymbol{z}^{\text{goal}}_t) \qquad \text{with} \quad \hat{r}_{t'} \sim o_\phi(\hat{r}_{t'} \mid \boldsymbol{s}_{t'}), \quad (22)$$

for a planning horizon $H$. The function $\text{sim}(\cdot)$ is a similarity measure between $\boldsymbol{z}^{\text{goal}}_t$ and $\boldsymbol{z}_t$. The hyperparameter $\kappa$ controls the trade-off between external and internal reward. Previously, similarity between Gaussian distributed $\boldsymbol{z}_t$ of the RSSM was estimated using cosine similarity (Mendonca et al., 2021). However, for the categorically distributed $\boldsymbol{z}_t$, the cosine similarity can be low even when they stem from the same distribution. Instead we use the cosine similarity of the logits, i.e.

$$\text{sim}(\boldsymbol{z}_t, \boldsymbol{z}^{\text{goal}}_t) = \frac{\boldsymbol{l}_t \cdot \boldsymbol{l}^{\text{goal}}_t}{\|\boldsymbol{l}_t\| \|\boldsymbol{l}^{\text{goal}}_t\|}, \quad (23)$$

where $\cdot$ is the dot product and $\boldsymbol{l}_t$ and $\boldsymbol{l}^{\text{goal}}_t$ are the logits of the distributions that produced $\boldsymbol{z}_t$ and $\boldsymbol{z}^{\text{goal}}_t$, respectively. Compared to other similarity measures, e.g. KL divergence, our measure has the desirable property that $\text{sim}(\boldsymbol{z}_t, \boldsymbol{z}^{\text{goal}}_t) \in [0, 1]$, which simplifies setting the hyperparameter $\kappa$, which we set to $\kappa = 0.025$ to mainly guide the behavior in the absence of external reward.

## 3 RESULTS

We empirically evaluate THICK to answer the following questions:

- **Can THICK learn temporal abstractions?** We show that the learned high-level world model indeed discerns meaningful, interpretable temporal abstractions across various scenarios (Sec. 3.1).
- **Can THICK's hierarchical predictions improve MBRL?** We show that THICK Dreamer achieves higher returns than Dreamer in long-horizon tasks with sparse rewards (Sec. 3.2).
- **Can THICK's world model be used to plan hierarchically?** We show that MPC with THICK world models is better than flat world models at solving long-horizon tasks (Sec. 3.3).

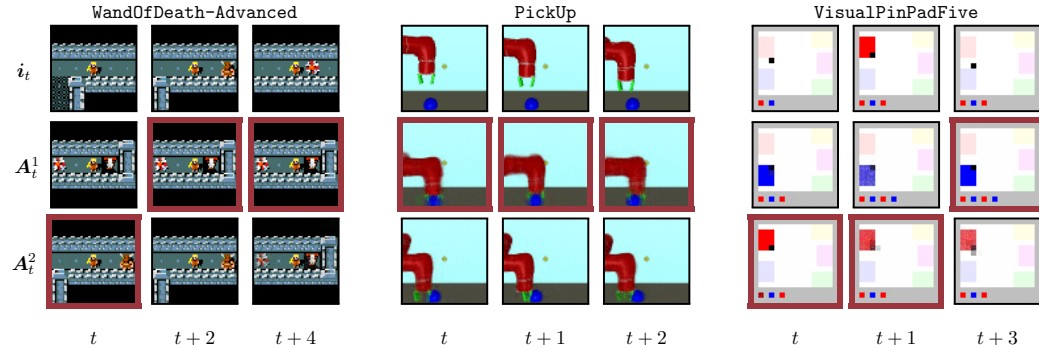

Figure 6: **High-level actions $A_t$.** We show input images $i_t$ and predictions $\hat{i}^c_{\tau(t)-1}$ for two high-level actions $A^1_t$ and $A^2_t$. Red frames depict sampled actions $\hat{A}_t$. Exemplar actions $A_t$ are: Exiting the room or attacking a monster (left), grasping or pushing a ball (center), activating pads (right).

We evaluate our THICK world models in various scenarios. **MiniHack** (Samvelyan et al., 2021) is a sandbox framework for designing RL environments based on Nethack (Küttler et al., 2020). We test our system on benchmark problems as well as newly created tasks. All problems, detailed in Suppl. E.1, have hierarchical structures in which subgoals need to be achieved (e.g. fetch a wand) to fulfill a task (e.g. kill a monster) to exit a dungeon and receive a sparse reward. The observation is a pixel-based, ego-centric view of $\pm 2$ grid-cells around the agent. MiniHack uses discrete actions.

**VisualPinPad** (Hafner et al., 2022) is a suite of visual, long-horizon RL problems. Here an agent (black square) needs to step on a fixed sequence of pads to receive a sparse reward. We use three levels of difficulties based on the number of pads and target sequence length (three, four, five).

**MultiWorld** (Pong et al., 2018) is a suite of robotic manipulation tasks for visual RL. In these tasks a Sawyer robot has to either move an object to a goal position (puck in `Pusher` or ball in `PickUp`) or open a door (`Door`). We use fixed goals and take the normalized distance between the to-be-controlled entity and the goal position as dense rewards (in `Pusher-Dense`, `PickUp`, `Door`) and thresholded distances as sparse rewards (in `Pusher-Sparse`). Details are provided in Suppl. E.2.

### 3.1 INTERPRETABLE CONTEXTS AND HIERARCHICAL PREDICTIONS

First, we analyze the predictions of THICK world models across diverse tasks. Example sequences are displayed in Fig. 5, in Suppl. F.1 and on our website. In MiniHack, context updates typically coincide with item collection, map changes, area exploration, or dungeon exits. In Multiworld, context changes occur due to object interactions or at workspace boundaries. In VisualPinPad, activating pads can prompt context changes. The high-level model predicts the states preceding context changes, often abstracting details, leading to blurry reconstructions. For instance, in `KeyRoom`, the system forecasts the agent's level exit without knowledge of the exact room layout (Fig. 5, $t + 6$). Nevertheless, the lower level consistently predicts the next frames accurately, as shown in Fig. 1b.

Abstract action representations $A_t$ emerge on the high level, as illustrated in Fig. 6. These actions categorically encode different agent-world interactions, e.g., grasping or pushing a ball in `PickUp`. The prior $Q_\theta$ learns to sample actions based on the likelihood of their outcomes (red frames in Fig. 6). If there are more actions $A_t$ than necessary, different actions encode the same outcome.

### 3.2 MODEL-BASED REINFORCEMENT LEARNING

We investigate whether hierarchical roll-outs can improve MBRL in the MiniHack suite by comparing THICK Dreamer to DreamerV2 (Hafner et al., 2020) and to Director (Hafner et al., 2022), a hierarchical RL method based on Dreamer. Fig. 7a–7d show that THICK Dreamer matches or outperforms flat Dreamer in all tasks in terms of sample efficiency or overall success rate. The advantage of THICK Dreamer is more pronounced in tasks that require completing multiple subgoals (e.g. completing five subgoals in `EscapeRoom` vs. finding a key to open a door in `KeyRoom`). Director outperforms the other methods in `KeyRoom` but fails to learn other MiniHack tasks. We investigate the failure cases of Director in Suppl. F.3 and show more MiniHack results in Suppl. F.2.

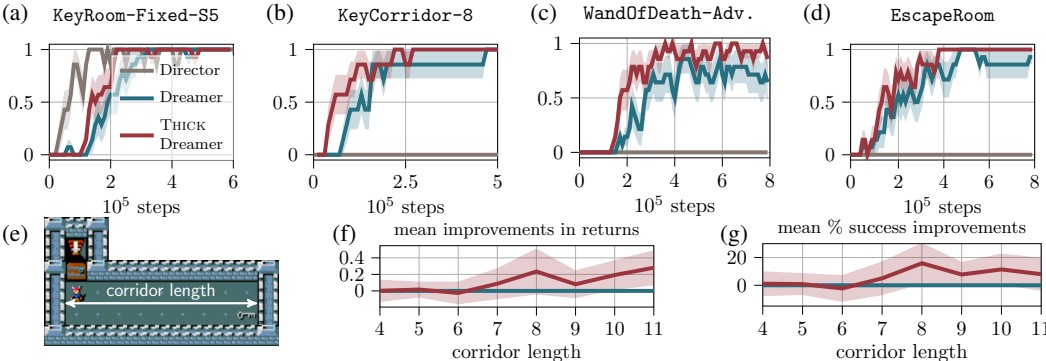

Figure 7: **MiniHack results**. Top graphics (a-d) plot the mean success rate during evaluation for various MiniHack tasks using 7 seeds. For `KeyCorridor` (e) we systematically vary corridor length and plot mean differences in evaluation returns (f) and percentage of task success (g) between THICK Dreamer and Dreamer over different lengths. Shaded areas depict ± one standard error.

We hypothesize that task horizon length is the main factor boosting THICK Dreamer's performance. To investigate this, we systematically vary the task horizon in the `KeyCorridor` problem (see Fig. 7e) by modifying the corridor length. Fig. 7f–7g plot the mean difference in obtained rewards and success rate over 500k steps of training between THICK Dreamer and Dreamer for different corridor lengths. The performance gain of THICK Dreamer tends to increase with corridor length until at some length both approaches fail to discover rewards during training, detailed in Suppl. F.2.

We further analyze the effect of task horizon in VisualPinPad. VisualPinPad poses two challenges: exploration and long-horizon behavior. To analyze the latter in isolation, we sidestep the challenge of discovering the sparse rewards by initially filling the replay buffer of all models with 1M steps of exploration using Plan2Explore (Sekar et al., 2020) (details in Suppl. F.4). Fig. 8 shows the performance of THICK Dreamer, DreamerV2, and Director. THICK Dreamer matches Dreamer in `PinPadThree` and is slightly more sample efficient in the more challenging tasks.[4] Thus, fusing hierarchical predictions to train a single policy in THICK Dreamer seems better suited for long-horizon learning than the hierarchical policies of Director or not employing hierarchies.

### 3.3 ZERO-SHOT MODEL-PREDICTIVE CONTROL

Lastly, we analyze whether our hierarchical predictions are suitable for planning by comparing THICK PlaNet and PlaNet (Hafner et al., 2019b) in Multiworld. We consider the challenging setup of MPC for models trained on an offline dataset of 1M samples collected by Plan2Explore (Sekar et al., 2020). Figure 9 shows the zero-shot performance over training. For `Pusher-Dense`, i.e. a short-horizon task[5] with dense rewards, there is no notable difference between both methods. When rewards are sparse (`Pusher-Sparse`) or the task horizon is long (`Door` and `PickUp`), THICK PlaNet achieves higher returns than PlaNet. Additionally, the subgoals set by the high level can be decoded, shown in Suppl. F.6, which improves the explainability of the system's behavior.

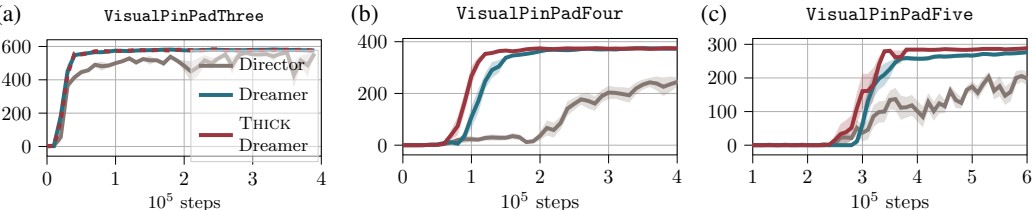

Figure 8: **VisualPinPad results**. We plot the mean evaluation returns for 7 seeds (± standard error).

---

[4]Previously, Hafner et al. (2022) reported that Director outperforms Dreamer in VisualPinPad. We hypothesize that this improvement stems from more sophisticated exploration, which is not necessary in our setting.

[5]Since the puck starts between the gripper and goal, the task can be solved by directly moving to the goal.

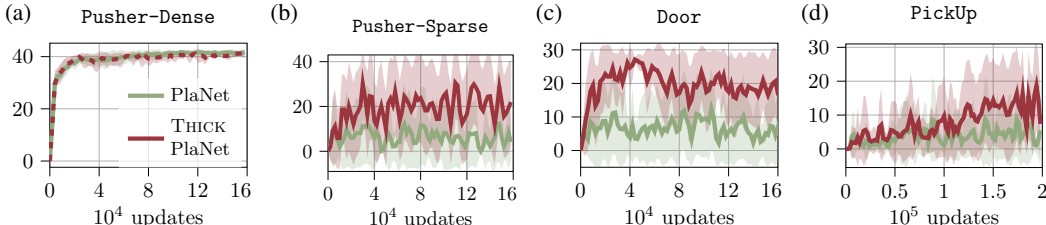

Figure 9: **MPC Multiworld results**. Each graphic plots the mean returns for zero-shot planning in Multiworld over world model updates using 10 seeds. Shaded areas depict the standard deviation.

## 4 RELATED WORK

**Sparsity in RNNs**: Learning hierarchical RNNs from sparse activity was proposed in Schmidhuber (1992), where a high level would become active based on low-level errors. Subsequently, there has been a lot of research on fostering sparsity in RNNs (Graves et al., 2014; Neil et al., 2016; Goyal et al., 2021; Gumbsch et al., 2021; Jain et al., 2022), which we compare in Suppl. C.

**Temporal abstract predictions**: One main challenge for learning temporal abstractions is segmenting a sequence into meaningful units. Discrete latent dynamics were previously used to model proactive gaze behavior (Gumbsch et al., 2022). Alternative segmentation methods are identifying easy-to-predict bottleneck states (Neitz et al., 2018; Jayaraman et al., 2019; Zakharov et al., 2021), using fixed time scales (Saxena et al., 2021), prediction error-based segmentation (Gumbsch et al., 2019), or regularizing boundary detectors (Kim et al., 2019; Zakharov et al., 2022) (details in Suppl. C).

**Hierarchical RL (HRL)**: HRL is an orthogonal research direction to hierarchical world models.In HRL a high-level policy either selects a low-level policy or provides goals or rewards for a low level (Pateria et al., 2021). In contrast, our THICK Dreamer uses high-level predictions to train a flat RL agent. Typically in HRL, the high level operates on fixed time scales (Hafner et al., 2022; Nachum et al., 2018; Vezhnevets et al., 2017; Gürtler et al., 2021) or task-dependently based on subgoal completion (Bacon et al., 2017; Levy et al., 2019). In THICK world models, the high level is learned time- and task-independently purely from predictions and latent state regularization.

## 5 CONCLUSION

We have introduced C-RSSM and THICK—fully self-supervised methods to construct hierarchical world models. By imposing a sparsity objective, C-RSSM develops context codes that update only at critical situations, where prediction-relevant aspects of the environment change. On a higher level, THICK learns to anticipate context-altering states. Categorical high-level action codes enable the anticipation of different outcomes, accounting for multiple lower-level context transitions. As a result, THICK world models can predict both abstract context transitions and exact low-level dynamics. Additionally, we have shown that the hierarchical predictions can improve long-horizon learning.

**Limitations**   THICK relies on setting the hyperparameter $\beta^{\mathrm{sparse}}$, which determines the high-level segmentation. Ideally, this hyperparameter should be tuned for every task. However, we found that the same value works well across similar tasks. Furthermore, except for improving long-horizon learning our downstream applications have similar restrictions as the method they build upon. For example, if Dreamer never discovers a solution to a task, THICK cannot decompose it.

**Future directions**   We see great potential of THICK world models as a tool to build more sophisticated agents that explore and plan their behavior across multiple time scales. A promising direction is combining MCTS with RL (Schrittwieser et al., 2020), e.g. for biologically plausible planning (Mattar & Lengyel, 2022) by searching for high-level goals that goal-condition low-level policies (Akakzia et al., 2021). Another potential lies in integrating more active epistemic-driven exploration (Sekar et al., 2020; Sancaktar et al., 2022), which could lead to a more robust consolidation of context codes and transitions between them. Future extensions could also explore richer predictions purely from the context $c_t$. This would allow the high-level to directly predict context transitions without predicting observable state information used for intermediate queries to the low-level. Lastly, while we employed THICK to establish a two-level hierarchy of world models, THICK could be applied on multiple levels to recursively build an $N$-level world model hierarchy.

## ACKNOWLEDGMENTS

We thank Karl Friston, Marco Bagatella, and Tomáš Daniš for the valuable feedback and helpful discussions. We acknowledge the support of the German Federal Ministry of Education and Research through the Tübingen AI Center (FKZ: 01IS18039B). This research was funded by the German Research Foundation (DFG) within Priority-Program SPP 2134 – project "Development of the agentive self" (BU 1335/11-1, EL 253/8-1). Georg Martius and Martin Butz are members of the Machine Learning Cluster of Excellence, EXC number 2064/1 – Project number 390727645. We thank the International Max Planck Research School for Intelligent Systems (IMPRS-IS) for supporting Christian Gumbsch. Noor Sajid is grateful for support from the Medical Research Council (MR/S502522/1) and the 2021-2022 Microsoft PhD Fellowship.

## REPRODUCIBILITY STATEMENT

We provide our source code on https://github.com/CognitiveModeling/THICK. Additionally, we specify all our hyperparameters choices, details on the conducted hyperparameter search, and advice on how to tune the hyperparameters for novel task in Suppl. B.

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

# Supplementary Material for:
# Learning Hierarchical World Models with Adaptive Temporal Abstractions from Discrete Latent Dynamics

**Contents**

# A    PSEUDOCODE

Algorithm 1 outlines how THICK world models make temporal abstract predictions using both levels of the hierarchy (also visualized in Fig. 4). Blue parts are only needed for MBRL or MPC (see Sec. 2.3). For temporal abstract rollouts, which are used in THICK PlaNet, the process can be repeated $K$ times by using the output states, i.e. $c_{\tau(t)}$ and $\hat{z}^c_{\tau(t)}$, as inputs again.

---

**Algorithm 1 THICK Temporal Abstract Prediction**

---

1: **input:** context $c_t$, stochastic state $z_t$
2: $\hat{A}_t \sim P_\theta(\hat{A}_t \mid c_t, z_t)$          ▷ sample high-level action
3: $\hat{z}_{\tau(t)-1} \sim F_\theta\big(\hat{z}_{\tau(t)-1} \mid \hat{A}_t c_t, z_t\big)$          ▷ high-level state prediction
4: $\hat{a}_{\tau(t)-1} \sim F_\theta\big(\hat{a}_{\tau(t)-1} \mid \hat{A}_t c_t, z_t\big)$          ▷ high-level action prediction
5: $\Delta\hat{\tau}(t) \sim F_\theta\big(\Delta\hat{\tau}(t) \mid \hat{A}_t c_t, z_t\big)$          ▷ high-level time prediction
6: $(\hat{r}^\gamma_{t:\tau(t)} \sim F_\theta\big(\hat{r}^\gamma_{t:\tau(t)} \mid \hat{A}_t c_t, z_t\big)$          ▷ high-level reward prediction
7: $c_{\tau(t)} \leftarrow g_\phi\big(\hat{a}_{\tau(t)-1}, c_t, \hat{z}_{\tau(t)-1}\big)$          ▷ low-level context
8: $\hat{z}^c_{\tau(t)} \sim p^c_\phi\Big(\hat{z}^c_{\tau(t)-1} \mid \hat{a}_{\tau(t)-1}, c_{\tau(t)}, \hat{z}_{\tau(t)-1}\Big)$          ▷ low-level coarse prior
9: $\hat{r}^c_{\tau(t)}, \hat{\gamma}^c_{\tau(t)} \sim o^c_\phi\Big(\hat{r}^c_{\tau(t)}, \hat{\gamma}^c_{\tau(t)} \mid c_{\tau(t)}, \hat{z}^c_{\tau(t)}\Big)$          ▷ coarse reward & discount prediction
10: **output:** $c_{\tau(t)}, \hat{z}^c_{\tau(t)}, \widehat{\Delta t}, \hat{r}^\gamma_{t_\tau}\ \hat{r}^c_{\tau(t)}, \hat{\gamma}^c_{\tau(t)}$

---

Algorithm 2 describes how to create input-target data for training the high-level world model. In continual learning environments with no early termination of an episode, we omit the red part.

---

**Algorithm 2 THICK Training Data Generation**

---

1: **input:** discount factor $\gamma$, sequences of contexts $c_{1:T}$, stochastic states $z_{1:T}$, actions $a_{1:T}$,
2:          rewards $r_{1:T}$, and episode termination flags $d_{1:T}$
3: **initialize:** train data $\mathcal{D} \leftarrow \{\}$, unassigned inputs $\mathcal{I} \leftarrow \{\}$
4: **for** $\tau \leftarrow 1$ to $T$ **do**
5:      **if** $c_\tau \neq c_{\tau-1}$ **or** $d_\tau = 1$ **then**          ▷ context change or episode is over at time $\tau$
6:          **for** $(c_t, z_t) \in \mathcal{I}$ **do**
7:              compute passed time $\Delta\tau \leftarrow \tau - t$ and accumulated rewards $r_{t:\tau} \leftarrow \sum_{\delta=1}^{\Delta t-1} \gamma^\delta r_{t+\delta}$
8:              add input-target tuple $\big((c_t, z_t), (z_{\tau-1}, a_{\tau-1}, \Delta t, r_{t:\tau})\big)$ to $\mathcal{D}$
9:              remove $(c_t, z_t)$ from $\mathcal{I}$
10:      add potential input $(c_\tau, z_\tau)$ to $\mathcal{I}$
11: **output:** train data $\mathcal{D}$

---

Algorithm 3 describes the general training and generation of behavior THICK world models. Red parts are only used for THICK PlaNet. Blue parts are only used for THICK Dreamer. In our zero-shot planning experiments using THICK PlaNet, we do not add new data to the replay buffer and only plan and execute actions during evaluation.

---

**Algorithm 3 THICK World Models**

---

1: initialize neural networks and replay buffer
2: $t^{\mathrm{plan}} = -I$
3: **for** $t \leftarrow 1$ to $t^{\mathrm{end}}$ **do**
4:     update low-level world model state $\boldsymbol{s}_t \sim w_\phi(\boldsymbol{s}_t \mid \boldsymbol{s}_{t-1}, \boldsymbol{a}_{t-1})$
5:     `// Behavior`
6:     **if** $\boldsymbol{c}_t \neq \boldsymbol{c}_{t-1} \wedge t \geq t^{\mathrm{plan}} + I$ **then**
7:         plan subgoal $\boldsymbol{z}_t^{\mathrm{goal}}$ using MCTS and temporal abstract rollouts (Alg. 1)
8:         $t^{\mathrm{plan}} \leftarrow t$
9:     plan new action $\boldsymbol{a}_t$ using CEM given $\boldsymbol{s}_t$ and $\boldsymbol{z}_t^{\mathrm{goal}}$ (Eq. 22)
10:     sample new action $\boldsymbol{a}_t$ from actor $\pi$ given $\boldsymbol{s}_t$
11:     execute action $\boldsymbol{a}_t$ in environment and observe $\boldsymbol{r}_t, \boldsymbol{i}_t$ and $d_t$
12:     add $(\boldsymbol{i}_t, \boldsymbol{a}_t, r_t, d_t)$ to replay buffer
13:     `// Train world models`
14:     draw sequence batch $\mathcal{B} \leftarrow (\boldsymbol{i}_{t':T}, \boldsymbol{a}_{t':T}, r_{t':T}, d_{t':T})$ from replay buffer
15:     embed batch in latent state $\boldsymbol{s}_{t':T} \sim w_\phi(\boldsymbol{s}_{t':T} \mid \mathcal{B})$
16:     update low-level world model $w_\phi$ using $\mathcal{B}$ (Eq. 9)
17:     generate high-level training batch $\mathcal{D}$ from $(\boldsymbol{s}_{t':T}, \boldsymbol{a}_{t':T}, r_{t':T}, d_{t':T})$ (Alg. 2)
18:     update high-level world model $W_\theta$ using $\mathcal{D}$ (Eq. 18)
19:     `// Train actor and critic`
20:     imagine trajectory $(\boldsymbol{s}_{t'':H}, \boldsymbol{a}_{t'':H}, r_{t'':H}, \gamma_{t'':H})$ using $w_\phi$ from random start $s_{t''} \in \mathcal{B}$
21:     make temporal abstract predictions for each $\boldsymbol{s}_{t'':H}$ using $W_\theta$ and $w_\phi$ (Alg. 1)
22:     compute value $V$ (Eq. 20)
23:     update critics $v_\chi$ and $v_\xi$ (Eq. 33)
24:     update actor $\pi$

---

## B  HYPERPARAMETERS

Table 1: **Hyperparameter choices.** If there is only one centered value it counts for all suites. Otherwise different values are chosen for MiniHack (MH), VisualPinPad (VPP), or Mulitworld (MW).

| Name | MH | VPP | MW |
|---|---|---|---|
| **Low-Level World Model** (C-RSSM or RSSM) | | | |
| Batches (size $\times$ sequence length) | | $16 \times 50$ | |
| Dimensions of $c_t$ | | 16 | |
| Dimensions of $h_t$ | | 256 | |
| Dimensions of $z_t$ | | $32 \times 32$ | |
| MLP features per layer | | 256 | |
| Sparsity loss scale $\beta^{\text{sparse}}$ | $1^6/10$ | 1 | 25 |
| Prediction loss scale $\beta^{\text{pred}}$ | | 1 | |
| KL loss scale $\beta^{\text{KL}}$ | | 1 | |
| KL balancing $\beta^{\text{bal}}$ | | 0.8 | |
| Output heads $o_\phi$ for | $i_t, \gamma_t, r_t$ | $i_t, r_t$ | $i_t, r_t$ |
| Prioritize ends in replay | yes | no | no |
| Learning rate | | 0.0001 | |
| **High-Level World Model** | | | |
| $Q_\theta$ & $P_\theta$ number of layers $\times$ features | | $3 \times 200$ | |
| $F_\theta$ number of layers $\times$ features | | $5 \times 1024$ | |
| Number of actions $A_t$ | 3 | 5 | 5 |
| Use terminations $d_t$ for segmentation | yes | no | no |
| Loss for training $F_\theta^{\hat{a}}\left(\hat{a}_{\tau(t)-1} \mid A_t, c_t, z_t\right)$ | CCE | CCE | NLL |
| Action prediction loss scale $\alpha^{a_{\tau(t)-1}}$ | 1 | 1 | 0.1 |
| State prediction loss scale $\alpha^{z_{\tau(t)-1}}$ | | 1 | |
| Time prediction loss scale $\alpha^{\Delta\tau(t)}$ | 1 | 1 | 0.1 |
| Reward prediction loss scale $\alpha^{r^\gamma_{t:\tau(t)}}$ | | 1 | |
| KL balancing $\alpha^{\text{bal}}$ | | 0.8 | |
| Learning rate | | 0.0001 | |
| **THICK Dreamer** | | | |
| Imagination horizon $H$ | 15 | 15 | |
| Value estimate balance $\psi$ | 0.9 | 0.9 | |
| $\lambda$-target of $V_t^\lambda$ | 0.95 | 0.95 | |
| Long-horizon critic $v_\chi$ layers $\times$ features | $4 \times 400$ | $4 \times 400$ | |
| Long-horizon critic $v_\chi$ learning rate | 0.0002 | 0.0002 | |
| **THICK PlaNet** | | | |
| CEM planning horizon $H$ | | | 12 |
| Long-horizon scale $\kappa$ | | | 0.025 |
| MCTS simulations | | | 100 |
| MCTS discount | | | 0.997 |
| **Common** | | | |
| Optimizer | | Adam | |
| MLP activation functions | | ELU | |
| Discount $\gamma$ | | 0.99 | |

---

[6]Unlike the other MiniHack problems, `MiniHack-Corridor` tasks are fully deterministic, which is why we use a lower factor of $\beta^{\text{sparse}} = 1$ here.

**World model learning hyperparameters**   For optimizing the world models, i.e. our THICK world models and the baseline models in Dreamer and Director, we use the default DreamerV2 hyperparameters (Hafner et al., 2020) – except for minor variations. Specifically, we reduced the model size by setting the feature size of the RSSM and the dimensionality of $h_t$ to 256. As we show in Suppl. F.7 model size does not strongly affect performance in our setting. Additionally, for THICK Dreamer and Dreamer we did not employ layer normalization for the GRU within the RSSM, because in pre-tests this showed increased robustness for both approaches.

**MBRL hyperparameters**   For training the actor and critic in THICK Dreamer and Dreamer, we use the default hyperparameters of DreamerV2. For Director we mostly used its default hyperparameters (Hafner et al., 2022), however, we made some minor adjustments to the training frequency to ensure a fair comparison. Director performs one training update every 16 policy steps instead of every 5 steps in DreamerV2. This was done to reduce wall-clock time but decreases sample efficiency (Hafner et al., 2022). We increase the update frequency ($16 \rightarrow 5$) in order to fairly compare sample efficiency between approaches.

**MPC hyperparameters**   For MPC with CEM we use the hyperparameters of PlaNet (Hafner et al., 2019b). For high-level planning with MCTS, we use MuZero's (Schrittwieser et al., 2020) implementation, with mostly the same hyperparameters. However, intuitively we would not expect multiple predictions to reach a goal. Thus, we decrease the number of simulations to $S = 100$.

**Differences between environments**   The main difference between MiniHack and the other environments is that in MiniHack episodes can terminate based on the success of the task or the death of the agent. VisualPinPad and Multiworld do not feature early episode termination. As is customary with DreamerV2, for environments that do not feature early episode termination, we do not predict discounts $\gamma_t$, nor do we prioritize the termination of episodes in the replay buffer. Importantly, we do not treat episode terminations as context changes. For action prediction, we use Categorical Cross Entropy Loss (CCE) for predicting discrete actions (Minihack and VisualPinPad), and scale down the high-level prediction loss for predicting actions and elapsed time when training purely on task-free offline data (Multiworld). Finally, the sparsity loss scale $\beta^{\text{sparse}}$ was tuned for each suite.

**Hyperparameter search**   For determining the sparsity loss scale $\beta^{\text{sparse}}$, the value estimate balance $\psi$, and the long-horizon planning scale $\kappa$, we ran a grid search using three random seeds and using two tasks of each suite (MiniHack: `KeyRoom-Fixed-S5`, `WandOfDeath-Advances`; MiniHack-Corridor: `KeyCorridor-4`, `KeyCorridor-8`, Visual Pin Pad: `VisuaLPinPadFour`, `VisuaLPinPadFive`; Multiworld: `Door`, `PickUp`). We determined the best hyperparameter value for each suite depending on task performance and a qualitative inspection of the high-level predictions (see Suppl. F.8). For simplicity and to demonstrate robustness, we used the same values for each suite, whenever it was reasonable.

**How to tune**   When tuning THICK world models for a new task, we recommend mainly searching over the sparsity loss scale $\beta^{\text{sparse}} \in \{1, 5, 10, 25, 50\}$. Typically, one random seed is sufficient to determine which $\beta^{\text{sparse}}$ leads to few, but not too few, context changes.

## C  EXTENDED RELATED WORK

**Sparsity in RNNs**: The development of hierarchical RNNs based on sparse activity was already proposed in the 90s by Jürgen Schmidhuber (Schmidhuber, 1992). The Neural History Compressor (Schmidhuber, 1992) uses a hierarchical stack of RNNs that autoregressively predict the next inputs. The higher levels in the hierarchy remain inactive until the lower level fails to predict the next input. Recently, there has been increasing interest in regularizing RNNs towards sparse latent updates. Alternative approaches to $L_0$ regularization of GateL0RD (Gumbsch et al., 2021) are to use sparse attention masks (Graves et al., 2014; Goyal et al., 2021), competition among submodules (Goyal et al., 2021), regularizing update gates towards a variational prior (Jain et al., 2022), or time-dependent updates (Koutnik et al., 2014; Neil et al., 2016).

**Temporal abstractions from regularized latent dynamics**: Previously, sparse changes in the latent states of a low-level model have been used to model temporal abstractions (Gumbsch et al., 2022; Saxena et al., 2021). In contrast to our work, the temporal abstractions in Gumbsch et al. (2022) were learned in much simpler settings with highly structured observations, instead of the high-dimensional, pixel-based observations examined in this work. Additionally, these temporal abstractions were only used to model goal-anticipatory gaze behavior of infants and have not been applied for MPC or MBRL. Separately, Saxena et al. (2021) introduced a hierarchical video prediction model (i.e. without action) that used different clock speeds at each level to learn long-term dependencies using pixel-based input. Although this was apt for learning slow-moving content at higher levels of the temporal hierarchy, unlike C-RSSM and THICK, it requires the temporal abstraction factor to be defined explicitly.

**Temporal abstractions from predictability**: Adaptive Skip Intervals (ASI) (Neitz et al., 2018) is a method for learning temporal abstract autoregressive predictions. In ASI, a network is trained to predict those inputs within a predefined horizon, that best allow predicting extended sequences into the future. As a result, the model learns to skip a number of inputs towards predictable transitions. Similarly, temporal-agnostic predictions (TAP) (Jayaraman et al., 2019) identify frames of a video within a time horizon that are highly predictable. TAP is then trained to only predict the predictable "bottleneck" frames. Zakharov et al. (2021) provide a learning-free mechanism to detect context change by evaluating how predictable future states are. Briefly, their approach detects changes in the latent representation of each layer in the model hierarchy and introduces temporal abstraction by blocking bottom-up information propagation between different contexts. This is different from THICK, where context changes are determined using a learning-based sparsity regularization. An opposing approach is to use unexpected prediction errors of a forward model for self-supervised time series segmentation (Gumbsch et al., 2019). Here, the idea is that in certain states, the dynamics of agent-environment interactions change, e.g. changing the terrain during locomotion, leading to a temporary increase in the prediction error.

**Temporal abstractions from learning boundary detectors**: In addition to using indirect measures to segment a sequence, a straightforward approach is to train a boundary detector that signals the boundary of subsequences (Kim et al., 2019; Zakharov et al., 2022). Kim et al. (2019) train a boundary detector that is regularized by specifying the maximum number of subsequences allowed and their maximum length. This requires prior knowledge about the training data and imposes hard constraints on the time scales of the learned temporal abstractions. Our sparsity loss instead implements a soft constraint. Conversely, Zakharov et al. (2022) introduced a boundary detection mechanism using a non-parametric posterior over the latent states. Here, the model learns to transition between states only if a change in the represented features had been observed – otherwise temporally persistent states were clustered together.

**Faster credit assignment in RL**: THICK Dreamer predicts long-horizon outcomes via its high-level model when training the critic in imagination to boost reward propagation. This allows for faster credit assignment for tasks with sparse or delayed rewards. Previously, this was tackled using reward redistribution (Patil et al., 2021).

# D  THICK WORLD MODELS: IMPLEMENTATION DETAILS

## D.1  THICK DESIGN CHOICES

Due to space constraints, we explain some design choices in more detail here.

**High-level targets**  Our goal is to learn a high-level world model that predict situations in which latent generative factors are assumed to change, e.g. a door openings or object manipulations. Besides that, we want to use high-level outputs to predict future rewards and reconstruct images at some time $\tau(t)$. Thus, we at least need the context $c_{\tau(t)}$ and the stochastic state $z_{\tau(t)}$ to make these reconstructions through the coarse processing pathway (see Eq. 8). There are two potential ways to predict $c_{\tau(t)}$ and $z_{\tau(t)}$, either by predicting the state *before* or *after* the context transition.

We predict the states *before* the context transition. Our main reasoning is that the prediction of context-altering situations presents two challenges: $i$) learning in which situation such transitions are currently possible and $ii$) how these transitions affect the latent generative factors. The C-RSSM already learns to encode $ii$). Thus, in order to reduce redundancy and simplify the challenge, we train the high-level model to learn only $i$) and then prompt the low-level model for $ii$). One example from MiniHack would be predicting the agent standing in front of closed door and performing a door-opening action. We believe that this is a simpler prediction compared to predicting the egocentric view of the agent after opening a door and looking into a (potentially unknown) new room.

**Coarse predictions for contextual learning**  We want the context $c_t$ to encode latent information that is necessary for prediction and reconstruction. If we omit the coarse prior predictions (Eq. 5) and coarse output reconstructions (Eq. 8) the C-RSSM would not have any incentive to encode prediction-relevant latent information in $c_t$. Instead, it could purely utilize $h_t$ and avoid a sparsity penalty in Eq. 9 via $\mathcal{L}^{\text{sparse}}$ by never updating $c_t$. Completely omitting $h$ in C-RSSM impedes learning, as we show in our ablations in Suppl. F.8. Thus, we instead add the coarse processing pathway. Through coarse predictions, C-RSSM must encode latent factors in $c_t$ in order to reduce the KL-loss (Eq. 31) and prediction loss (Eq. 30).

**Coarse predictions to omit $h_t$**  The high-level model attempts to predict a future state of the system. The full latent state would contain the deterministic component $h$. However, for the high-level model it would be very challenging to predict the unregularized high-dimensional deterministic hidden state $h$ many time steps in the future. The coarse pathway of the C-RSSM allows to update the context dynamics $c_t$, predict stochastic states $z_t^c$, and reconstruct external variables without the deterministic hidden state $h_t$. Thus, it is advantageous that the C-RSSM can make predictions without $h$. After a high-level prediction, we can feed the high-level outputs $(\hat{z}_{\tau(t)-1}, \hat{a}_{\tau(t)-1})$ into the low-level world model. This brings many advantages: for example, this allows us to predict rewards or discounts/episode termination at particular states after a high-level prediction, which we use in THICK Dreamer in and THICK PlaNet (see Sec. 2.3). Furthermore, we can reconstruct images to visualize predictions as shown in Sec. 3.1. Additionally, we can continue with low-level rollouts after a high-level prediction, which is a feature that we have not yet utilized.

## D.2  GATEL0RD

We want the context code $c_t$ to change only sparsely over time. Thus, we implement the discrete context dynamics $g_\phi$ as a GateL0RD cell (Gumbsch et al., 2021). GateL0RD is an RNN designed to maintain sparsely changing latent states $c_t$. To realize this inductive bias, GateL0RD uses two subnetworks $g_\phi^p$ and $g_\phi^g$ that control $c_t$-updates via an internal update gate $\Lambda_t$. GateL0RD can be summarized as follows:

$$\text{Candidate proposal:}\quad \hat{c}_t = g_\phi^p(a_{t-1}, c_{t-1}, z_{t-1}) \tag{24}$$

$$\text{Update gate:}\quad \Lambda_t = g_\phi^g(a_{t-1}, c_{t-1}, z_{t-1}) \tag{25}$$

$$\text{Context Update:}\quad c_t = \Lambda_t \circ \hat{c}_t + (1 - \Lambda_t) \circ c_{t-1} \tag{26}$$

with $\circ$ denoting the Hadamard product. We use the action $a_{t-1}$ and the last stochastic state $z_{t-1}$ as the cell inputs. Based on this cell input and the last context $c_{t-1}$, GateL0RD proposes a new context

$\hat{c}_t$ via its proposal subnetwork $g_\phi^p$ (Eq. 24). Whether the context is updated depends on an update gate $\mathbf{\Lambda}_t \in [0,1]^m$ (Eq. 26). This update gate $\mathbf{\Lambda}_t$ is the output of the gating subnetwork $g_\phi^g$ (Eq. 25) which uses a rectified $\tanh$ activation function (ReTanh), with $\mathrm{ReTanh}(x) := \max(0, \tanh(x))$. This ensures that the gate activations are $\in [0,1]^m$. Note that to compute $\mathbf{\Lambda}_t$, the subnetwork $g_\phi^g$ internally samples from a Gaussian distribution before applying the ReTanh function. This was shown to improve robustness (Gumbsch et al., 2021). Thus, context updates are a stochastic process.

Originally (Gumbsch et al., 2021), GateL0RD used a subnetwork to compute the cell output using multiplicative gating. We omit this here and instead feed the context directly to the GRU cell $f_\phi$ as shown in Fig. 2 (right).

The centralized gate $\mathbf{\Lambda}_t$ of GateL0RD makes it easy to determine the changes in context, i.e. $\mathbf{c}_t \neq \mathbf{c}_{t-1}$. Since all the context updates depend on $\mathbf{\Lambda}_t$, we know that the context changed if $\mathbf{\Lambda}_t > \mathbf{0}$. This is an advantage over other RNNs that use multiple gates for sparsely changing latent states. We use this measure to determine context changes when building the world model hierarchy.

### D.3  C-Rssm Loss

The loss of the C-RSSM (Eq. 9) is composed of three parts: the prediction loss $\mathcal{L}^{\mathrm{pred}}$, the KL loss $\mathcal{L}^{\mathrm{KL}}$, and the sparsity loss $\mathcal{L}^{\mathrm{sparse}}$. Except for the sparsity loss, we adapt these loss terms from the RSSM. However, we always need to account for the coarse prediction pathways of the C-RSSM.

We define the prediction loss $\mathcal{L}^{\mathrm{pred}}$ as

$$\mathcal{L}^{\mathrm{pred}}(\phi) = \frac{1}{T}\sum_{t=1}^{T}\Big[\sum_{y \in \{\mathbf{i}_t, \mathbf{r}_t, \gamma_t\}} -\log o_\phi(y \mid \mathbf{s}_t) - \log o_\phi^c(y \mid \mathbf{c}_t, \mathbf{z}_t)\Big]. \tag{27}$$

Equations in red are exclusive to the C-RSSM. Thus, the network is trained to minimize the negative log likelihood for predicting the images $\mathbf{i}_t$, rewards $r_t$ and future discounts $\gamma_t$. Here we account for both the precise predictions over the output heads $o_\phi$ (Eq. 7), as well as for the coarse predictions over the output heads $o_\phi^c$ (Eq. 8). Following the codebase of DreamerV2 (Hafner et al., 2020), in continual learning environments when there is no early episode termination, we do not predict the discount $\gamma_t$, and instead use a fixed discount $\gamma = 0.99$.

The C-RSSM predicts two prior distributions for the next stochastic state $\hat{\mathbf{z}}_t$: fine predictions using the full state (Eq. 4) and coarse predictions based only on the context, last action, and stochastic state (Eq. 5). We need to account for both types of prediction in the KL loss $\mathcal{L}^{\mathrm{KL}}$ with

$$\mathcal{L}^{\mathrm{KL}}(\phi) = \frac{1}{T}\sum_{t=1}^{T}\mathrm{KL}\Big[q_\phi\big(\mathbf{z}_t \mid \mathbf{h}_t, \mathbf{i}_t\big)||p_\phi^h\big(\hat{\mathbf{z}}_t \mid \mathbf{h}_t\big)\Big] + \mathrm{KL}\Big[q_\phi\big(\mathbf{z}_t \mid \mathbf{h}_t, \mathbf{i}_t\big)||p_\phi^c\big(\hat{\mathbf{z}}_t^c \mid \mathbf{a}_{t-1}, \mathbf{c}_t, \mathbf{z}_{t-1}\big)\Big]. \tag{28}$$

Thus, we want to minimize the divergence between the fine prior $p_\phi^h$ and the approximate posterior $q_\phi$, as well as the divergence between the coarse prior $p_\phi^c$ and $q_\phi$. As in DreamerV2 (Hafner et al., 2020), we use KL-balancing, which scales the prior $p_\phi$ of each KL divergence by a factor $\beta^{\mathrm{bal}} = 0.8$, and the posterior $q_\phi$ by $1 - \beta^{\mathrm{bal}}$. This enables faster learning of the prior to avoid that the posterior is regularized towards an untrained prior.

We take the sparsity loss $\mathcal{L}^{\mathrm{sparse}}$ from GateL0RD (Gumbsch et al., 2021), which is a $L_0$-regularization of the context changes $\mathbf{\Delta c}_t$. This is implemented as

$$\mathcal{L}^{\mathrm{sparse}}(\phi) = \frac{1}{TJ}\sum_{t=1}^{T}\sum_{j=1}^{J}\Big\|\Delta c_t^j\Big\|_0 = \frac{1}{TJ}\sum_{t=1}^{T}\sum_{j=1}^{J}\Theta\big(\Lambda_t^j\big) \tag{29}$$

where $J$ is the dimensionality of the context $\mathbf{c}_t$ and $\Theta(\cdot)$ denotes the Heaviside step function. That is, an $L_0$-regularization of the context changes is implemented as the binarization of the update gates $\mathbf{\Lambda}_t$ (Eq. 26). We estimate the gradient of the Heaviside step function using the straight-through estimator (Bengio et al., 2013). The advantage of GateL0RD's $L_0$-regularization to other regularization toward

sparsity, such as using low variational prior (Jain et al., 2022), is that in fully-observable and highly predictable situations, GateL0RD will shut its gates and not change the context almost regardless of the order of magnitude of $\beta^{\text{sparse}}$ to avoid a punishment.

## D.4 HIGH-LEVEL WORLD MODEL TRAINING

The high-level world model with parameters $\theta$ is trained to minimize both the prediction loss $\mathfrak{L}^{\text{pred}}$, of predicting the next context change state, and the action loss $\mathfrak{L}^{\boldsymbol{A}}$, which is the divergence of prior and posterior high-level action distributions.

For every high-level target $Y \in \{\boldsymbol{a}_{\tau(t)-1}, \boldsymbol{z}_{\tau(t)-1}, \Delta\tau(t), r_{t:\tau(t)}^{\gamma}\}$, we use an appropriate prediction loss and employ a weighted sum in $\mathfrak{L}^{\text{pred}}$. For the to-be-predicted action $\boldsymbol{a}_{\tau(t)-1}$, passed time $\Delta\tau(t)$, and rewards $r_{t:\tau(t)}^{\gamma}$, we use the negative log-likelihood (NLL). For stochastic state prediction $\boldsymbol{z}_{\tau(t)-1}$, we know the underlying distribution that generated the target variable, i.e. the posterior $q_{\phi}(\cdot)$. Thus, we can use the KL divergence as a loss for high-level state predictions. Overall, we get

$$\mathfrak{L}^{\text{pred}}(\theta) = \frac{1}{T} \sum_{t=1}^{T} \Big[ \sum_{Y \in \{\boldsymbol{a}_{\tau(t)-1}, \Delta\tau(t), r_{t:\tau(t)}^{\gamma}\}} -\alpha^{Y} \log F_{\theta}(Y \mid \boldsymbol{A}_t, \boldsymbol{c}_t, \boldsymbol{z}_t)$$
$$+ \alpha^{\text{z}} \text{KL} \Big[ q_{\phi}\big(\boldsymbol{z}_{\tau(t)-1} \mid \boldsymbol{c}_{\tau(t)-1}, \boldsymbol{h}_{\tau(t)-1}, \boldsymbol{i}_{\tau(t)-1}\big) || F_{\theta}^{\hat{\boldsymbol{z}}}\big(\hat{\boldsymbol{z}}_{\tau(t)-1} \mid \boldsymbol{A}_t, \boldsymbol{c}_t, \boldsymbol{z}_t\big) \Big] \Big].$$
(30)

The hyperparameters $\alpha^{Y} \in \{\alpha^{\boldsymbol{a}}, \alpha^{\boldsymbol{z}}, \alpha^{\Delta\tau(t)}, \alpha^{r^{\gamma}}\}$ can be used to scale the individual loss terms. By default, we set $\alpha^{Y} = 1$ for all loss terms. When training the network on task-free exploration, i.e. during zero-shot MPC as described in Sec. 3.3, we found that predicting actions $\boldsymbol{a}_{\tau(t)-1}$ at context changes and elapsed time $\Delta\tau(t)$ was challenging. To mitigate this, during task-free exploration we set $\alpha^{\text{a}_{\tau(t)-1}} = 0.1$ and $\alpha^{\Delta\tau(t)} = 0.1$. For predicting continuous actions we sample from a Gaussian distribution of predicted actions and compute the NLL as the loss for action prediction. For discrete actions we predict a Categorical distribution from which we sample the actions, and compute the Categorical Cross Entropy (CCE) loss.

The action loss $\mathfrak{L}^{\boldsymbol{A}}$ drives the system to minimize the divergence between the posterior high-level action distribution $Q_{\theta}(\boldsymbol{A}_t \mid \boldsymbol{c}_t, \boldsymbol{z}_t, \boldsymbol{c}_{\tau(t)}, \boldsymbol{z}_{\tau(t)})$, and the prior distribution $P_{\theta}(\hat{\boldsymbol{A}}_t \mid \boldsymbol{c}_t, \boldsymbol{z}_t)$ with

$$\mathfrak{L}^{\boldsymbol{A}}(\theta) = \frac{1}{T} \sum_{t=1}^{T} \text{KL} \Big[ Q_{\theta}(\boldsymbol{A}_t \mid \boldsymbol{c}_t, \boldsymbol{z}_t, \boldsymbol{c}_{\tau(t)}, \boldsymbol{z}_{\tau(t)}) \quad || \quad P_{\theta}(\hat{\boldsymbol{A}}_t \mid \boldsymbol{c}_t, \boldsymbol{z}_t) \Big].$$
(31)

Like the KL loss $\mathcal{L}^{\text{KL}}$ on the low level (see Suppl. D.3), we use KL balancing (Hafner et al., 2020) to scale the prior part by $\alpha^{\text{bal}} = 0.8$ and the posterior part by $1 - \alpha^{\text{bal}}$.

## D.5 THICK DREAMER: DETAILS

THICK Dreamer estimates the overall value $V(\boldsymbol{s}_t)$ of a state $\boldsymbol{s}_t$ as a mixture of short- and long-horizon estimates (Eq. 20). The short-horizon value estimate is computed as the general $\lambda$-target as in DreamerV2 with

$$V^{\lambda}(\boldsymbol{s}_t) = \hat{r}_t + \hat{\gamma}_t \begin{cases} (1-\lambda)v_{\xi}(\hat{\boldsymbol{s}}_{t+1}) + \lambda V^{\lambda}(\hat{\boldsymbol{s}}_{t+1}) & \text{for } t < H, \\ v_{\xi}(\hat{\boldsymbol{s}}_{t+1}) & \text{for } t = H \end{cases}$$
(32)

where $\hat{r}_t$ and $\hat{\gamma}_t$ are sampled from the output heads $o_{\phi}$ given $\boldsymbol{s}_t$ (Eq. 7) and $\lambda$ is a hyperparameter.

THICK Dreamer trains both critics $v_{\xi}$ and $v_{\chi}$ to regress the overall value estimate using a squared loss:
$$\mathcal{L}(\vartheta) = \mathbb{E}_{p_{\phi}} \Big[ \sum_{t=1}^{H} \frac{1}{2} \big( v_{\vartheta}(\boldsymbol{s}_t) - \text{sg}\big(V(\boldsymbol{s}_t)\big) \big)^2 \Big],$$
(33)

for the two critics $v_{\vartheta} \in \{v_{\xi}, v_{\chi}\}$ with parameters $\vartheta \in \{\xi, \chi\}$, and $\text{sg}(\cdot)$ the stop gradient operator.

The functions $V^{\lambda}$ and $V^{\text{long}}$ compute value targets using the critics $v_{\xi}$ and $v_{\chi}$, respectively. Like DreamerV2, we stabilize critic training by using a copy of the critics during value estimation (in Eq. 32 and Eq. 19). The copy is updated every 100 updates.

### D.6 THICK PLANET: DETAILS

For planning at the high level, we use an MCTS implementation based on MuZero (Schrittwieser et al., 2020). We only replan at the high level if the context changes, i.e. $c_t \neq c_{t-1}$. Since all subgoals $z_t^{\text{goal}}$ are situations that lead to context changes, no additional criterion for subgoal completion is needed. Upon reaching a subgoal, e.g. touching an object, the context can sometimes change for multiple subsequent time steps. This causes the high-level to replan multiple times in a row. To avoid a high computational load from replanning and to enable smoother trajectories, we inhibit replanning for $I = 3$ time steps after setting a new subgoal. While this could potentially degrade performance in dynamic environments, we found this to work well in Multiworld.

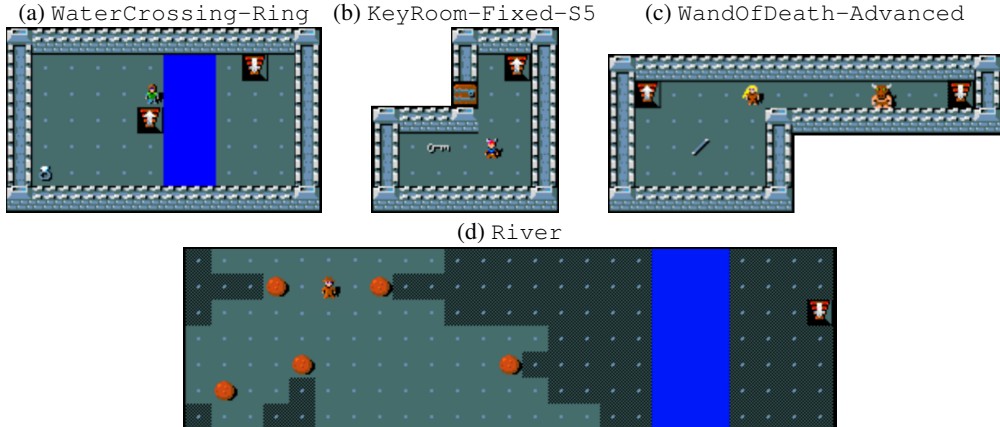

(a) `WaterCrossing-Ring`  (b) `KeyRoom-Fixed-S5`  (c) `WandOfDeath-Advanced`

(d) `River`

Figure 10: **MiniHack environments**. Staircases with an upward facing arrows mark the starting point of the agents. Staircases with downward facing arrows are the exits that need to be reached. In (a), (b), and (d) start points and exits are randomized.

## E   ENVIRONMENT DETAILS

### E.1   MINIHACK

Here we provide a detailed explanation of all the MiniHack problems we considered. In all settings, we restricted the action space to the minimum number of actions needed to solve the task. In all tasks, the agents receive a sparse reward of 1 when exiting the room and a small punishment of $-0.01$ when performing an action that has no effect, e.g. moving against a wall. In easier tasks (`WaterCrossing-Ring`, `KeyRoom-Fixed-S5`, `KeyCorridor`) the agent is allowed 200 time steps to solve the task. In all other tasks, the time limit is set to 400 time steps. For aesthetic reasons, we use different characters in different levels.

`WaterCrossing-Ring` is a newly designed, simple level in which an agent needs to fetch a randomly placed ring of levitation and float over a river to get to the goal (Fig. 10a). When a ring is picked up in our tasks, it is automatically worn[7]. The level is inspired by `LavaCross-Levitate-Ring-PickUp` from the MiniHack benchmark suite, where a river of deadly lava blocks the exit. However, we found that Dreamer struggles to learn this task, because of the early terminations when entering the lava.

`KeyRoom-Fixed-S5` is a benchmark task, in which an agent spawns in a room at a random position and has to fetch a randomly placed key to open a door and enter a smaller room with a randomly located exit (Fig. 10b). The door position is fixed. In all our tasks, using the key opens the door from any grid cell adjacent to the door, even diagonally.

`KeyCorridor-N` is a novel task, in which an agent starts in front of a locked door in the top left corner of a 2-grid-wide corridor. In the bottom right corner of the corridor is the key for the door. We vary the length `N` of the corridor to systematically manipulate the task horizon.

`WandOfDeath-Advanced` is based on the `WandOfDeath` benchmark tasks, in which an exit is guarded by a minotaur, which instantly kills the agent upon contact. The agent needs to pick up a wand to attack and kill the monster. Thus, the agent needs to carefully select the direction of the attack, because if the attack bounces off a wall, it kills the agent instead. `WandOfDeath` comes in multiple levels of difficulty. `WandOfDeath-Advanced` (Fig. 10c) is a self-created level layout, designed to be more challenging than `WandOfDeath-Medium` but not as difficult as `WandOfDeath-Hard`. In `WandOfDeath-Medium` the agent can only walk horizontally and the location of the wand is fixed. In `WandOfDeath-Hard` the map is very large, making this a hard

---

[7]Usually, to wear a ring in MiniHack a sequence of actions needs to be performed: PUTON → RING → RIGHT, for putting the ring on the right finger. We simplify this by automatically applying the action sequence when the ring is picked up.

(a) `Pusher`         (b) `Door`         (c) `PickUp`

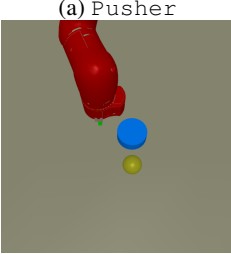 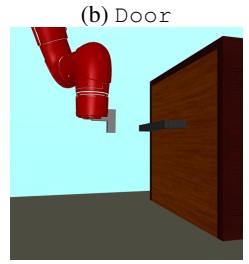 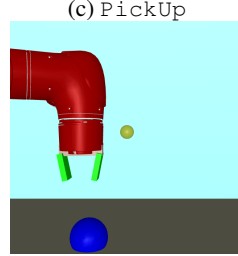

Figure 11: **Multiworld environments**. Goal positions for the objects are shown in yellow.

exploration problem. Our version is of intermediate difficulty, where the number of accessible grids (28) is roughly the same as in `WandOfDeath-Medium` (27), while the randomly placed wand needs to be found first. The minotur is asleep as in `WandOfDeath-Medium`.

`River` is a benchmark task, in which an agent needs get to an exit on the other side of a river (Fig. 10d). In order to cross the river the agent needs to push boulders into the water to form a small land bridge. To solve the task the agent needs to move at least two randomly placed boulders into the river.

`EscapeRoom` is a difficult new problem designed by us, which combines the challenges of many other problems (Fig. 14a). Using `EscapeRoom` we test the ability to learn to execute a complex event sequence of five subgoals. Nonetheless, the task can be learned without extensive exploration or large action spaces. The agent starts in a small room and the goal is to unlock a door and escape. However, to get the key, the agent must (1.) pick up a ring of levitation and (2.) float over a small patch of water into a corridor. In the corridor, the agent can (3.) exchange the ring of levitation for a key. In order to get back to the door in the first room, the agent needs to (4.) push a boulder into water. Finally, the agent can (5.) unlock the door and exit the room. During levitation, the agent is too light to push the boulder. In `EscapeRoom`, the agent can only carry one new item and picking up a second item results in dropping the first one.

### E.2  MULTIWORLD

In Multiworld we use tasks that have previously been used to study visual reinforcement learning (Nair et al., 2018; Pong et al., 2020). All tasks in Multiworld use different action spaces and camera viewpoints for their pixel-based observation, shown in Fig. 11. In `Pusher` the 2-dimensional actions control the $x-$ and $y-$movement of the endeffector, whereas the gripper is fixed. In `Door` the robot has a hook instead of a gripper at its endeffector and the 3-dimensional action controls $x-$, $y-$, and $z-$movement. In `PickUp` the 3-dimensional action controls the $y-$ and $z-$movement and the gripper opening. We binarized the gripper opening to prevent accidental object drops. In all tasks, the goal positions are fixed. In `Pusher` and `PickUp` they are visible in the video frames. In `Door` the goal is to open the door fully. For `Pusher-Dense` and `PickUp` we compute the reward $r_t$ for every time step $t$ as

$$r_t = 1 - \frac{\delta_t}{\delta_1}, \tag{34}$$

where $\delta_t$ is the Euclidean distance between object and goal at time $t$. For `Pusher-Sparse` the agent received a reward of $r_t = 1$ when the euclidean distance between the puck and the goal $\delta_t < 0.025$, otherwise $r_t = 0$. For `Door` the reward $r_t$ is the current angle of the door joint.

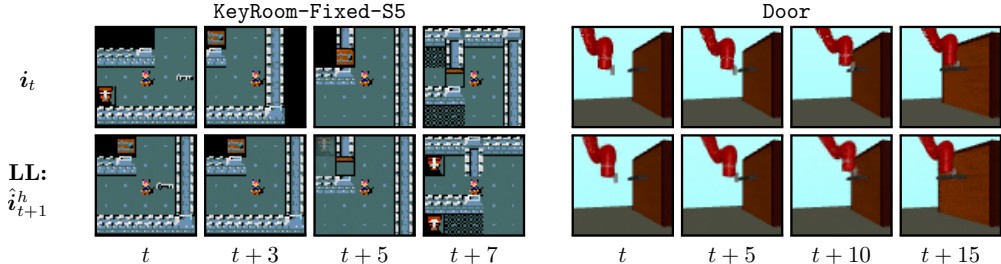

Figure 12: **Low-level predictions**. The low-level predictions accurately predict the next frames (cf. Fig. 5 for the high-level predictions and contexts $c_t$ of the same sequences).

## F EXTENDED RESULTS AND EXPERIMENT DETAILS

### F.1 ANALYSIS OF CONTEXTS AND PREDICTIONS

In this section, we provide further examples of high- and low-level predictions and context codes $c_t$.

**C-RSSM predictions and contexts** Figure 12 visualizes the low-level predictions for two example sequences. The low-level world model predicts the immediate next state and the reconstructions are more accurate than the abstract high-level predictions (cf. Fig. 5). Figure 13 displays four example sequences with the corresponding contexts $c_t$ and high-level predictions.

**High-level actions** We analyze the high-level actions $A_t$ in more detail for the EscapeRoom problem. EscapeRoom is a challenging MiniHack level, designed to contain diverse agent-environment interactions, shown in Fig. 14a and described in detail in Suppl. E.1. To illustrate the emerging high-level action representations of THICK Dreamer, we show inputs $i_t$ and image recon-

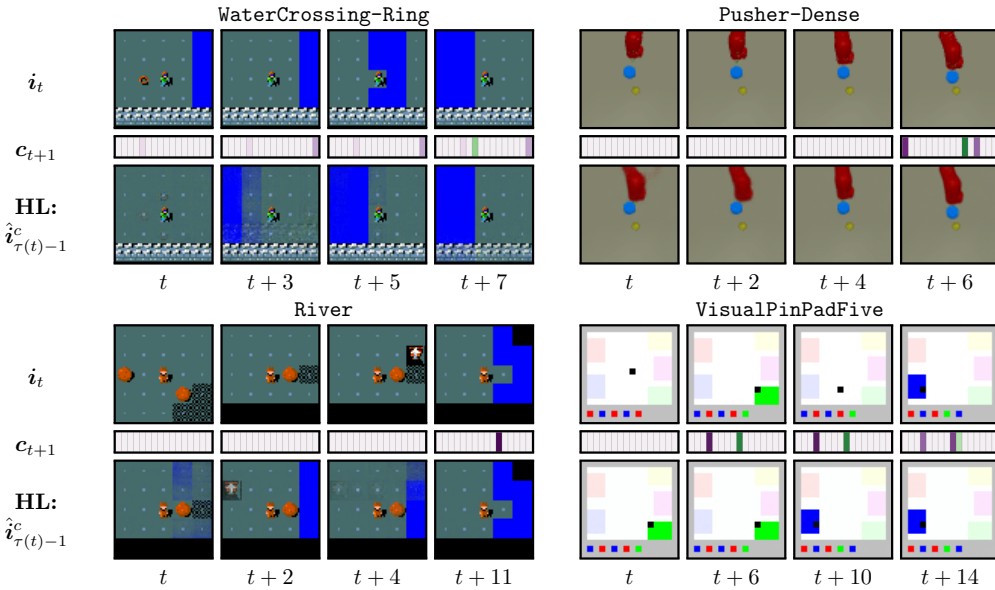

Figure 13: **Context changes and high-level predictions**. We show the input images $i_t$, 16-dim. contexts $c_{t+1}$ and reconstructions $\hat{i}^c_{\tau(t)-1}$ of high-level predictions. For WaterCrossing-Ring the context changes when stepping on the ring, picking it up, or arriving on the other side of the shore. In Pusher the context changes when the robot moves the puck. In River the context changes when pushing a boulder into water. In VisualPinPadFive the context changes when stepping on a pad.

(a)                                                    (b)

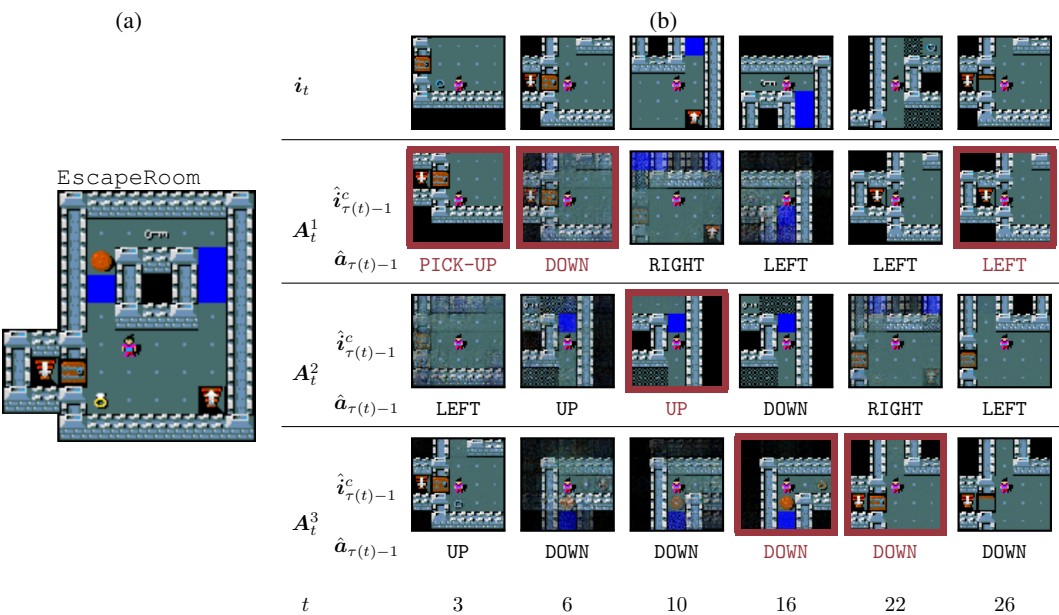

Figure 14: **High-level action predictions.** (a) In the `EscapeRoom` problem an agent needs to pick up a ring of levitation, hover over a patch of water to get to a key, exchange the ring for key, push a boulder into the water, and use the key to unlock a door. (b) Visualization of high-level actions for one exemplary sequence. The upper row shows the input image $i_t$. Image reconstructions $\hat{i}^c_{\tau(t)-1}$ and low-level action predictions $\hat{a}_{\tau(t)-1}$ are shown for the three high-level actions $A_t$. Red outlines depict which action $\hat{A}_t$ was sampled.

structions $\hat{i}^c_{\tau(t)-1}$ and predicted low-level actions $\hat{a}_{\tau(t)-1}$ for all high-level actions $A_t$ in Fig. 14b for one exemplary sequence.

At specific time steps, the three possible high-level actions $A_t$ encode particular agent-environment interactions: $A^1_t$ encodes picking up the ring of levitation ($t = 3$) or exiting the level ($t \in \{22, 26\}$). $A^2_t$ encodes crossing the water after obtaining the ability to levitate ($t \in \{6, 10, 16\}$), either upward ($t \in \{6, 10\}$) or downward ($t = 16$). $A^3_t$ encodes pushing the boulder into water ($t \in \{6, 10, 16\}$) or stepping in front of the door ($t = 22$). For all other time steps, high-level actions produce either identity predictions (e.g. $A^1_6$) or predictions that seem to encode average scene settings (cf. $A^2_3$ or $A^1_{10}$). These predictions account for unexpected context shifts, which can always occur with a small probability due to the stochasticity from sampling $z_t$ and the stochastic update gates of GateL0RD (see Suppl. D.2). The low-level actions predicted for these situations seem to be mostly random. The

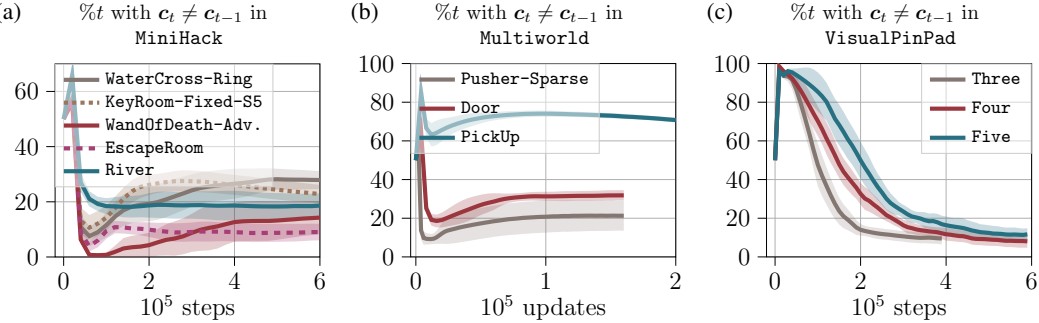

Figure 15: **Context changes over training**. Each graphic plots the mean percentage of time steps per training batch for which the context $c_t$ changes in MiniHack (a), VisualPinPad (b), and Multiworld (c). Shaded areas depict the standard deviation.

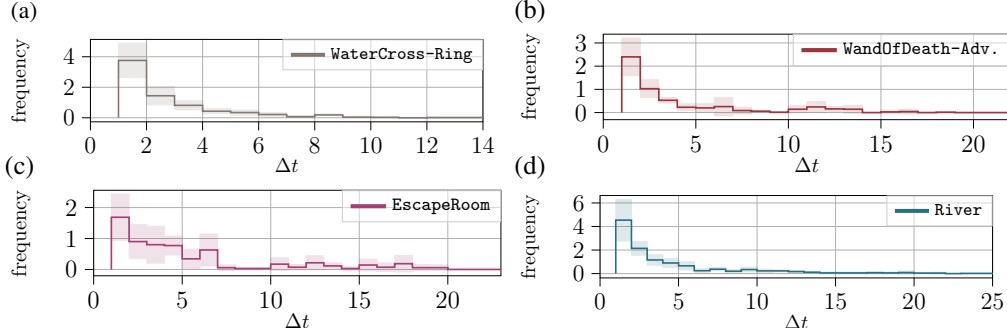

Figure 16: **Context duration during an episode**. Each graphic plots a histogram of the mean number of time steps $\Delta t$ between two consecutive context changes during an episode ( over 10 episodes, max 50 steps) for different MiniHack tasks (7 seeds). Shaded areas depict the standard deviation.

prior $Q_\phi$ (red frames and text in Fig. 14b) typically samples reasonable high-level actions. However, occasionally the prior samples an action $\hat{A}_t$ that leads to an identity or an average prediction (e.g. $t = 6$) due to the randomness of the process.

**Quantifying context changes**   To quantify the changes in context, we plot the mean percentage of time steps when context changes occur (i.e. $c_t \neq c_{t-1}$) over the course of training in Fig. 15. Importantly, context changes are somewhat consistent within the same task, but, as expected, can vary between tasks in the same suite despite using the same hyperparameter $\beta^{\text{sparse}}$. Additionally, we analyze the time between context changes for some MiniHack tasks. We plot the histogram of time gaps between context changes in Fig. 16 which illustrates that different tasks also show different distributions of context durations.

**Task-relevance of contexts**   Lastly, we analyze whether context changes occur in task-relevant situations for some MiniHack problems. For this, we generate rollouts using the fully trained policy and identify points $t^*$, which we consider to be crucial for solving the task. For `WandOfDeath-Adv.`, `WaterCross-Ring`, and `KeyRoom-Fixed-S5` we take the time points $t^*$ before picking up an item. For `EscapeRoom`, we use points in time $t^*$ when the agent stands in front of a movable boulder blocking the exit path. We compute the mean percentage of context changes occurring around $t^*$ ($\pm 1$ step) over 10 sequences and take the average over all 7 randomly seeded models. The results are shown in Table 2. The C-RSSM tends to update its context with a high probability in the identified situations. This suggests that task-relevant aspects, such as item pickups or boulder pushes, are encoded in the contexts.

Table 2: **Task-relevance of context changes**. We list the mean percentage of context changes $c_{t^*} \neq c_{t^*-1}$ for fully trained policies (7 seeds) at crucial task relevant points in time $t^*$ in 10 sequences. See text for criterion of $t^*$. The standard deviation is denoted by $\pm$.

|   | WaterCross | KeyRoom-Fixed-S5 | WandOfDeath-Adv. | EscapeRoom |
|---|---|---|---|---|
| % | 97.1 ($\pm$ 4.9) | 91.4 ($\pm$ 6.9) | 91.4 ($\pm$ 1.5) | 88.57 ($\pm$ 15.7) |

### F.2   MBRL IN MINIHACK: EXPERIMENT DETAILS AND EXTENDED RESULTS

In Fig. 17 we plot the success rate of THICK Dreamer, DreamerV2, and Director for additional MiniHack tasks not shown in the main paper.

To investigate the effect of task horizon, we compare the performance gain of THICK Dreamer over Dreamer for different corridor lengths in `KeyCorridor`. To compute improvements for every seed, we subtract the success rate and returns of Dreamer of THICK Dreamer (visualized in Fig. 7f–7g). For corridor lengths of 6 onward, the improvements of THICK Dreamer over Dreamer tend

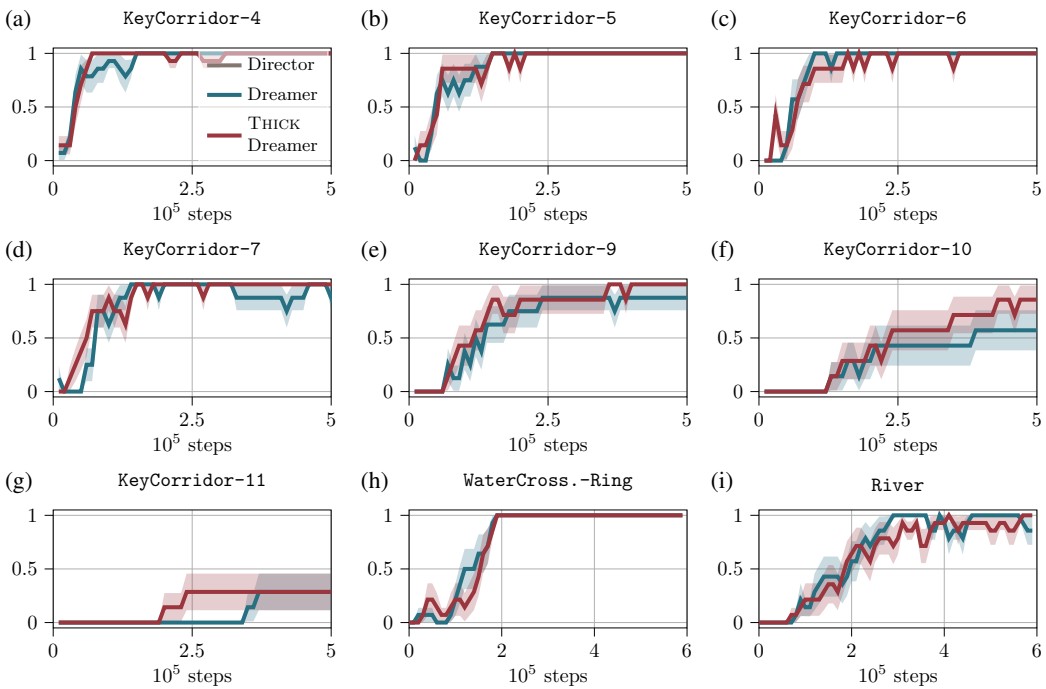

Figure 17: **MiniHack success**. We plot the mean evaluation success rate (7 seeds, $\pm$ standard error).

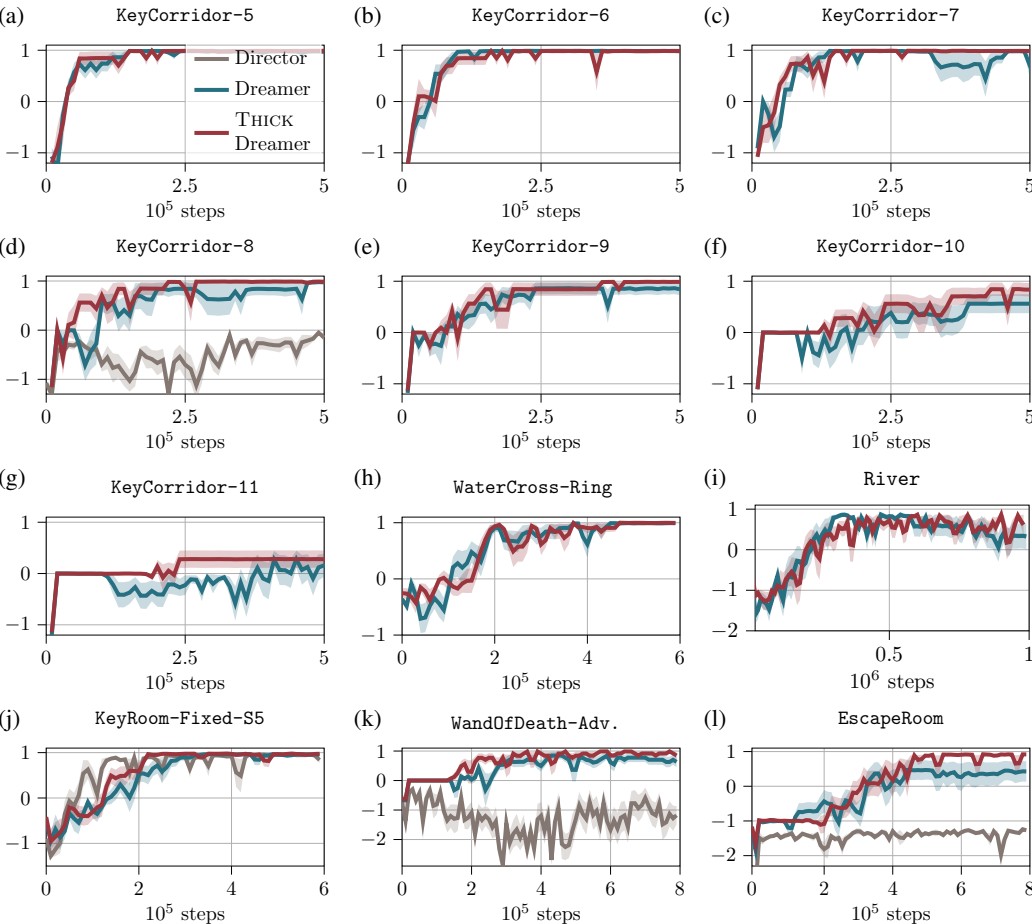

Figure 18: **MiniHack returns**. We plot the mean evaluation returns (7 seeds, $\pm$ standard error).

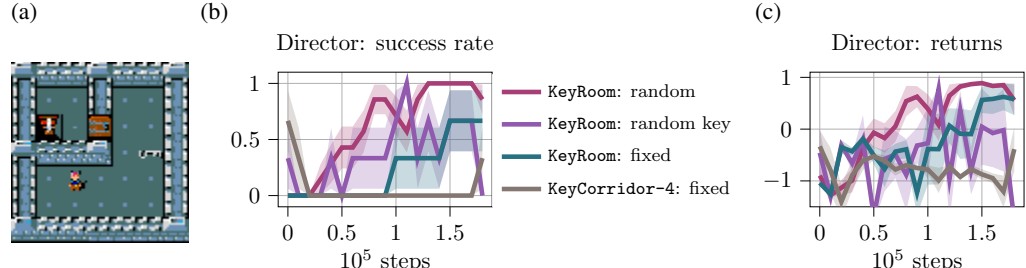

Figure 19: **Director in KeyDoor environments**. We test Director in variants of the `KeyRoom-Fixed-S5` problem, with fixed spawn positions of agent, key, and goal (i.e. `KeyRoom`: fixed, shown in a), random spawn points of the key (i.e. `KeyRoom`: random key), fully randomized spawn points (`KeyRoom`: random) or the similar `KeyCorridor-4` problem. We plot the mean evaluation success rate (b) and the mean evaluation returns (c) (7 seeds for `KeyRoom`: random, 3 seeds otherwise, $\pm$ standard error).

to increase with corridor length. However, for a corridor length of 11 most runs fail to discover the reward (see Fig. 17g), which dampens the improvement in performance. The inability to solve these tasks seems to come from inadequate exploration. Our results in VisualPinPad indicate that if exploration is addressed, then the performance gain of THICK Dreamer also holds for longer task horizons.

## F.3 MBRL IN MINIHACK: DIRECTOR

Director (Hafner et al., 2022) shows strong performance in the `KeyRoom-Fixed-S5` task. However, Director does not learn the other MiniHack tasks that we considered Director (Fig. 7). Which crucial aspects are different across tasks and what causes Director to fail? We identify two key problems when applying Director in MiniHack, namely $i$) Director's **reliance on diverse initial data** and $ii$) problems with **specifying unobservable goal information**.

**Diversity of initial data**    Director trains a goal encoder on its replay buffer from which it samples when training a goal-conditioned policy. We hypothesize that if early in training not enough diverse data is collected, this is reflected in the goal space. As a result, the manager (high-level) does not set meaningful goals for the worker (low-level) and learning is severely slowed down.

We analyze this aspect by training Director on variants of the `KeyRoom-Fixed-S5` problem. By default, the initial positions of the agent, key, and goal within the second room are randomized in `KeyRoom-Fixed-S5`. We create additional variants of the task in which all entities are spawned in fixed positions (positions shown in Fig. 19a) or only the initial position of the key is randomized. Additionally, we train Director in the `KeyCorridor-4` task, which is very similar to `KeyRoom-Fixed-S5` with fixed spawn points but of much smaller size (8 grid corridor vs. two rooms with 16 and 4 grids). Thus, in `KeyCorridor-4` the observations show little diversity.

Figure 19b plots evaluation success rates of Director in the `KeyRoom`-variants. Director needs more steps to solve the tasks when entities spawn at fixed positions. Director does not learn to solve `KeyCorridor-4` whereas with the same training it consistently learns to solve `KeyRoom-Fixed-S5`. Note that, `KeyCorridor-4` is much smaller and has a shorter task horizon. A similar trend can be observed in the collected returns (Fig. 19c).

Thus, we conclude that diversity in the initial observations drastically boosts Director's performance. The ego-centric views of MiniHack often contain the same or similar observations, especially when traversing long corridors or empty rooms, e.g. in `KeyCorridor-8` or `WandOfDeath-Advanced`. This similarity in observations might impede Director's learning in the MiniHack tasks we considered here.

**Unobservable aspects of goals**    We hypothesize that a severe problem of Director could be specifying unobservable information in the goals. The RSSM encodes both observable and unobservable information within its deterministic latent state $h_t$. If the unobservable information, e.g. item pick-

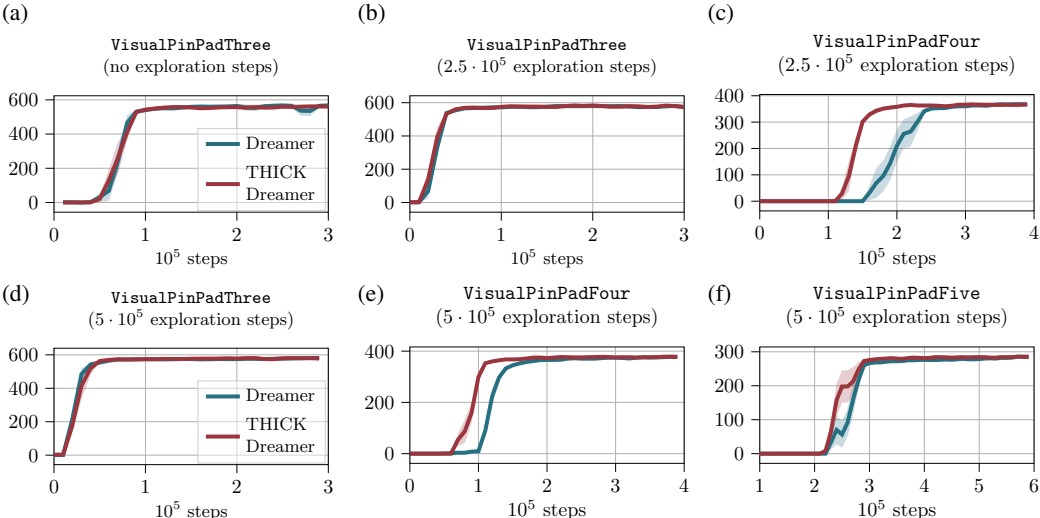

Figure 20: **VisualPinPad results and exploration**. We plot the mean evaluation returns (7 seeds for $5 \cdot 10^5$ exploration, 5 seeds otherwise; $\pm$ standard error).

ups in MiniHack, does not directly affect rewards or substantially influence image reconstruction, it might be encoded only locally in $h_t$ and wash out over time. In Dreamer this is not a problem because the policy can amplify task-relevant information in $h_t$. Director, however, compresses $h_t$ into a low-dimensional goal-encoding. Thereby, task-relevant latent information could get lost. Note that all novel tasks proposed in Hafner et al. (2022), in which Director shows very strong performance, avoid long-horizon memory, e.g. by coloring walls in a maze (Egocentric Ant Maze) or by providing a visible history of past button presses (VisualPinPad).

In smaller MiniHack tasks, e.g. KeyRoom-Fixed-S5, memory can sometimes be circumvented by specifying goals via observable information. For example, if both the key and door are visible, a goal would be the same observation without the key visible (picked up) and an open door. This creates a curriculum in which the worker can first learn from such simpler situations and later learn to pick up a key and open the door automatically from the high-level goal of an open door. In larger task spaces, e.g. KeyCorridor-8, Director never encounters such simpler situations to begin with.

### F.4 MBRL IN VISUALPINPAD: EXPERIMENT DETAILS

For the VisualPinPad suite we generated offline training data to bypass the challenge of discovering very sparse rewards. For data collection, we used Plan2Explore (Sekar et al., 2020) with the default settings of the DreamerV2 (Hafner et al., 2020) codebase. We trained two randomly initialized models of Plan2Explore for $S \in \{0, 250k, 500k, 1M\}$ environment steps in each task of the Visual Pin Pad suite. For each setting, we determined the model that achieved the highest overall returns during training. We initialized the replay buffer of all new models with the $S$ samples.

Originally, VisualPinPad has more levels of difficulty. However, in VisualPinPadSix Plan2Explore did not receive any reward during 1M steps of exploration. In addition to that, the results in Hafner et al. (2022) suggest that Dreamer is also not able to discover the very sparse rewards of VisualPinPadSix on its own. Thus, we omitted VisualPinPadSix and more complicated levels.

### F.5 MBRL IN VISUALPINPAD: EFFECT OF EXPLORATION

We analyze the effect of exploration data by varying the number of data points with which we initialize the replay buffers. For this, we consider exploration data collected by $S \in \{0, 250k, 500k\}$ environment steps of Plan2Explore and compare THICK Dreamer to Dreamer. In PinPadThree, Dreamer and THICK Dreamer always achieve the same performance regardless of exploration data

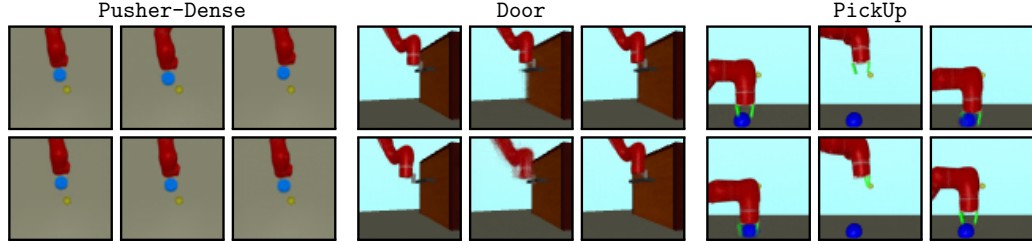

Figure 21: **Subgoals proposed in the first time step**. We reconstruct images based on the subgoals $z_1^{\text{goal}}$ that THICK PlaNet set at the first time step. The subgoals are typically pushing the puck (`Pusher`), moving to the door handle (`Door`), or grasping the ball (`PickUp`). For `PickUp` the system sometimes fails to find a reasonable subgoal (center).

available (cf. Fig. 20a, Fig. 20b, Fig. 20d). Without exploration data, neither THICK Dreamer nor Dreamer manage to obtain rewards in `PinPadFour` and `PinPadFive` within 600k steps. Similarly, both methods do not discover rewards when initialized with 250k steps of exploration in `PinPadFive`. Thus, for the more complicated problems, both THICK Dreamer and Dreamer need sufficient exploration. Whenever there is enough exploration data to learn the more complicated tasks, THICK Dreamer manages to achieve high rewards faster than Dreamer (see Fig. 20c, Fig. 20e, Fig. 20f).

For the larger problems, i.e. `PinPadFour` and `PinPadFive`, we quantify the effect of exploration data on sample efficiency by determining the number of environment steps needed to reach a certain level of reward. We take $95\%$ of the highest mean reward in all of our experiments as a threshold. This corresponds to a mean reward of 359 for `PinPadFour` and 274 for `PinPadFive`. Table 3 shows the number of environment steps needed for THICK Dreamer and Dreamer reach this threshold for particular environments over the amount of exploration data. In sum, a medium amount of exploration data (500k) enables reaching the threshold the fastest. THICK Dreamer reaches the reward threshold faster than Dreamer in all experiments. This advantage increases with level difficulty.

Table 3: **Sample efficiency in Visual Pin Pad**. We list the number of environment steps needed for THICK Dreamer, Dreamer, and their difference to pass a reward threshold (95% of max. reward) during evaluation for particular environments and amount of exploration data.

|  | PinPadFour | | |
| --- | --- | --- | --- |
| exploration data | 250k | 500k | 1M |
| THICK Dreamer | 200k | 120k | 140k |
| Dreamer | 280k | 180k | 200k |
| difference | 80k | 60k | 60k |

|  | PinPadFive | | |
| --- | --- | --- | --- |
| exploration data | 250k | 500k | 1M |
| THICK Dreamer | / | 260k | 340k |
| Dreamer | / | 360k | 590k |
| difference | / | 100k | 250k |

## F.6 MPC: EXPERIMENT DETAILS AND EXTENDED RESULTS

To study zero-shot planning, we generated offline datasets of 1M environment steps for every task. For data collection, we used Plan2Explore in the same way as described in Suppl. F.4. After determining one dataset for each task, we trained the models purely on this data.

In addition to increasing performance for long-range tasks, THICK PlaNet provides the additional advantage that the subgoals proposed by the high-level network can be directly reconstructed into

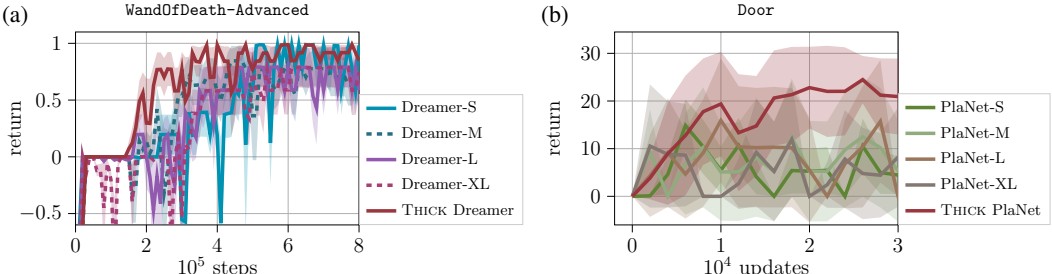

Figure 22: **Effect of scaling RSSM size**. We modify the number of hidden units per layer and dimensionality of the deterministic hidden state $h_t$ by a factor $(S = 0.5, M = 1, L = 2, XL = 4)$ for Dreamer (a) or PlaNet (b) and compare to the unmodified THICK counterparts. Each graphic plots the mean returns over training steps (5 random seeds for ablations). Shaded areas depict the standard error in (a) or standard deviation in (b).

images through the low-level output heads $o_\phi^c$. The resulting goal images are easily interpretable by humans. Figure 21 shows exemplary goals selected by the high-level planner in the first time step of an episode. Thus, the behavior of THICK PlaNet is much more explainable than simply performing MPC in a flat world model.

### F.7 MBRL & MPC: SCALING MODELS VS. HIERARCHY

An alternative approach to adding hierarchies to world models is to improve performance by scaling up the model size (Hafner et al., 2023; Deng et al., 2023). Could simply increasing the capacity of the world model improve performance of Dreamer or PlaNet similarly to our approach of incorporating hierarchical abstraction?

We investigate this in the `WandOfDeath-Adv.` task for MBRL and in `Door` for zero-shot MPC. We increase the RSSM model capacity by scaling up the number of hidden units per layer (256 before) and the dimensionality of the determinstic hidden state $h_t$ (256 before) by different factors (factors: $S = 0.5, M = 1, L = 2, XL = 4$). Unlike the model scaling in Hafner et al. (2023), we did not increase the dimensionality of $z_t$ as both investigated environments are visually rather simple. Figure 22 plots the returns of the different model sizes over environment steps. In both tasks, increasing the model size did not improve Dreamer (Fig. 22a) or PlaNet (Fig. 22b). Thus, for the investigated setups, scaling up model size does not bring the same advantages as our THICK hierarchy.

### F.8 ABLATIONS AND HYPERPARAMETERS

**Ablations** We ablate various components of the C-RSSM and THICK Dreamer within a MBRL setup. We evaluate the resulting systems using the two exemplary tasks of `MiniHack-WandOfDeath-Advances` and `VisualPinPadFour`. Figure 23 plots the returns of the ablated systems over environment steps. Using the C-RSSM in DreamerV2 results in roughly the same performance (`WandOfDeath-Advances`) or slightly better performance (`VisualPinPadFour`) than using the RSSM (i.e. Dreamer). However, removing the deterministic latent state $h_t$ and the precise processing pathway from the C-RSSM (i.e. C-RSSM Dreamer without $h$) impedes the system from learning the tasks.[8] Omitting $v_\xi$, and only using one critic $v_\chi$ for both the short- and long-horizon returns (Eq. 20), slightly degrades the performance of THICK Dreamer.

**Hyperparameter $\beta^{\mathrm{sparse}}$** Next, we compare the effect of sparsity loss scale $\beta^{\mathrm{sparse}}$ on THICK Dreamer in `VisualPinPadFour` and on THICK PlaNet in `Multiworld-Door`.

---

[8]For this ablation we picked higher sparsity regularization $\beta^{\mathrm{sparse}}$ for both tasks ($\beta^{\mathrm{sparse}} = 50$ for `WandOfDeath-Advances`, $\beta^{\mathrm{sparse}} = 10$ for `VisualPinPadFour`), such that the number of time steps with open gate roughly matches that of the C-RSSM.

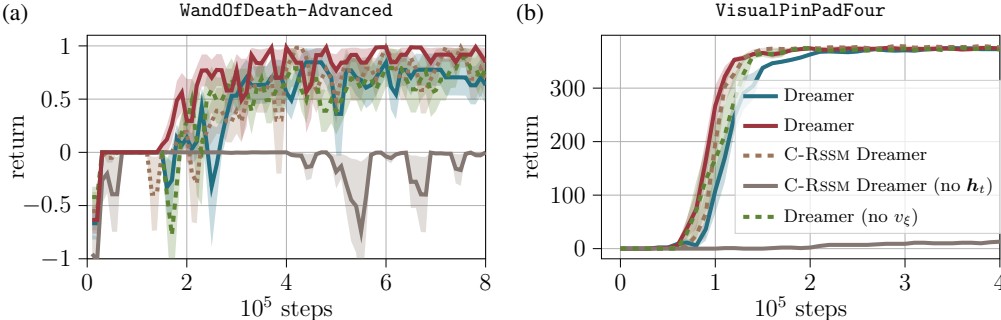

Figure 23: **C-RSSM and THICK Dreamer ablations**. Each graphic plots the mean returns over training steps. We compare Dreamer and THICK Dreamer (7 random seeds) against various ablations (5 seeds each): Dreamer using the C-RSSM (C-RSSM Dreamer), Dreamer using only the coarse processing pathway of the C-RSSM (C-RSSM Dreamer no $h_t$), and THICK Dreamer using only one critic (THICK Dreamer no $v_\xi$). Shaded areas depict the standard error.

Figure 24a plots the mean returns of THICK Dreamer or different values for $\beta^{\text{sparse}}$ in `VisualPinPadFour`. Figure 24b shows the percentage of time steps with context changes over training. For THICK Dreamer, regularizing context changes too little is not as detrimental as overly regularizing context changes. If the contexts are weakly regularized, i.e. small $\beta^{\text{sparse}}$, then the context changes in most time steps. As a result, the high-level learns an identity mapping, and during a temporal abstract prediction, the network simply predicts the next state at time $t+1$ (see Alg. 1). Stronger regularization boosts sample efficiency of learning long-horizon behavior. This is even true if, at some point after the behavior has been sufficiently learned, the context is no longer adapted (e.g. $\beta^{\text{sparse}} = 10$). However, overly strong regularization, which prohibits context changes early during training, impedes the learning of the task (e.g. $\beta^{\text{sparse}} = 100$). In this case, the high-level predictions are essentially average state predictions, which simply contributes noisy values for learning the critic. THICK Dreamer is very robust to the choice of $\beta^{\text{sparse}}$ in `VisualPinPadFour`.

Figure 24c plots zero-shot planning performance of THICK PlaNet for different values for $\beta^{\text{sparse}}$, with the percentage of context changes shown in Fig. 24d. For THICK PlaNet both too strong and too weak regularization degrade performance. However, strongly regularizing the network toward sparse context changes is slightly less detrimental for THICK PlaNet than a weak sparsity regularization (cf. $\beta^{\text{sparse}} = 100$ and $\beta^{\text{sparse}} = 5$). For weak sparsity regularization, the context changes in every time step, which prevents the high level from finding a useful subgoal sequence during planning. As a result, the low-level might be guided into the wrong direction by the proposed subgoals.

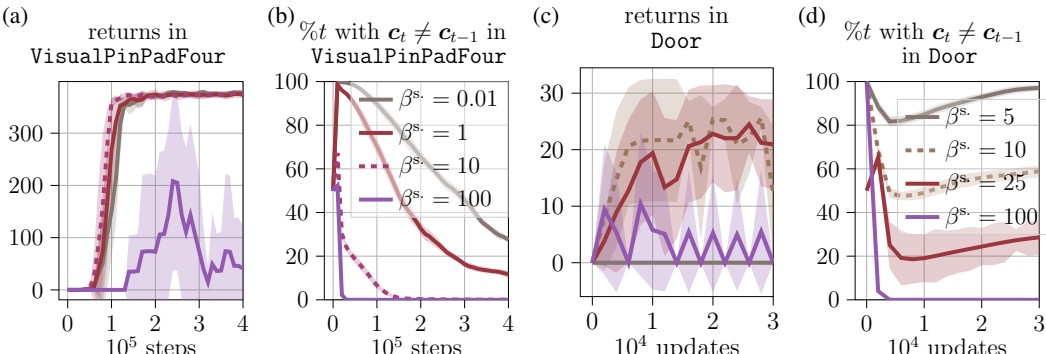

Figure 24: **Effect of sparsity**. We plot the mean return for THICK Dreamer in `VisualPinPadFour` (a) and the mean zero-shot planning returns of THICK PlaNet in `Door` (c) for different values of the hyperparameter $\beta^{\text{sparse}}$ (5 random seeds). In addition, we plot the percentage of time steps with context changes over training time for both tasks (b, d). We test different ranges of $\beta^{\text{sparse}}$ for the different tasks. Shaded areas depict the standard deviation.

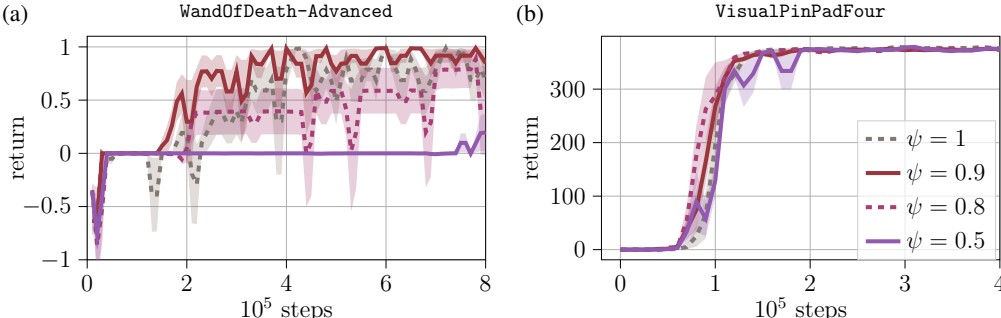

Figure 25: **Effect of hyperparameter** $\psi$. Each graphic plots the mean returns for THICK Dreamer over training steps for different values of the hyperparameter $\psi$ (5 random seeds). Shaded areas depict the standard error.

**Hyperparameter** $\psi$   THICK Dreamer introduces a new hyperparameter $\psi$ which balances the influence of the short-horizon value estimates $V^\lambda$ and long-horizon value estimates $V^{\text{long}}$ on the overall value $V$ (Eq. 20). Figure 25 shows how $\psi$ affects task performance. Only considering short-horizon value estimates, i.e. $\psi = 1$, results in less sample efficient learning than taking small amounts of long-horizon value estimates into consideration, i.e. $\psi = 0.9$ for WandOfDeath-Advances and $0.8 \leq \psi \leq 0.9$ for VisualPinPadFour. However, relying too strongly on long-horizon estimates, i.e. $\psi = 0.5$, impedes policy learning. This effect is less pronounced for very long-horizon tasks such as VisualPinPadFour. We set $\psi = 0.9$ in all experiments.

**Hyperparameter** $\kappa$   THICK PlaNet introduces the hyperparameter $\kappa$, which scales the influence of the subgoal proximity on the reward estimate of the low-level planner (Eq. 22). We analyze the effect of $\kappa$ on THICK PlaNet's performance in Multiworld-Door, shown in Fig. 26. Incentivizing subgoal proximity too strongly, i.e. $\kappa = 1$, can result in the agent getting stuck at a subgoal. This reduces overall task performance. Ignoring the subgoal, i.e. $\kappa = 0$, also decreases performance for long-horizon tasks such as Door. In Door, THICK PlaNet works well across a wide range of $\kappa$.

**Replanning strategy**   THICK PlaNet proposes new goals on the high-level upon context transitions. We do this mainly to save computational cost from running MCTS at the high-level at every time step. Figure 27 compares the effect of high-level planning in every time step to replanning upon context transitions in Door. The returns seem mostly the same. Thus, replanning at every time step is as effective as setting new subgoals only upon context transitions and can be applied if computational efficiency is not a concern.

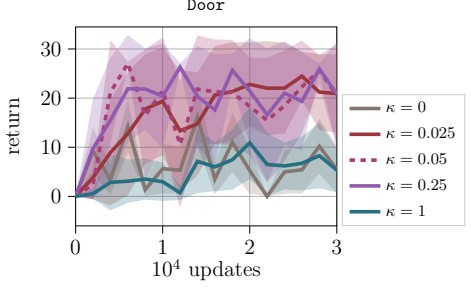
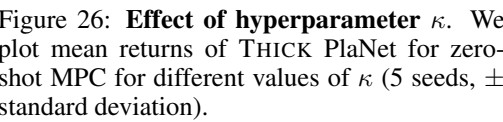
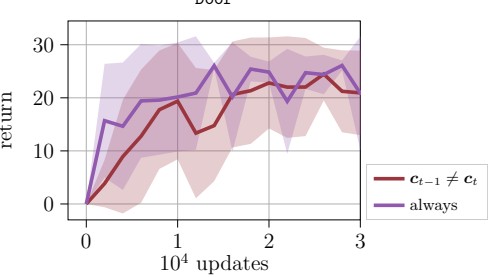

Figure 26: **Effect of hyperparameter** $\kappa$. We plot mean returns of THICK PlaNet for zero-shot MPC for different values of $\kappa$ (5 seeds, $\pm$ standard deviation).

Figure 27: **Effect of replanning**. We plot mean returns of THICK PlaNet for zero-shot MPC when replanning upon context transition or every step (5 seeds, $\pm$ standard deviation).

## G   COMPUTATION AND CODE

All experiments were run on an internal computing cluster. Each experiment used one GPU. Experiments using DreamerV2 took roughly 15-20 hours, while THICK Dreamer experiments took around 35-45 hours of wall clock time, depending on the overall number of environment steps. Experiments with Director took around 40-60 hours of wall clock time. Zero-shot MPC experiments with PlaNet took about 30-35 hours for $10^6$ training steps, while THICK PlaNet took roughly 50-60 hours for the same number of training steps. The higher wall clock time for training THICK world models stems mainly from the greater number of trainable parameters and more detailed logging. In addition to that, THICK PlaNet takes longer to evaluate, due to the additional computational cost of running MCTS on the high level during planning. The code for running the experiments is available online.

