# OpenReview forum: "Learning Hierarchical World Models with Adaptive Temporal Abstractions from Discrete Latent Dynamics"
_ICLR.cc/2024/Conference — ICLR 2024 spotlight_

### Official Review · Reviewer_yLBk · 2023-11-01

**Soundness:** 4 excellent
**Presentation:** 4 excellent
**Contribution:** 3 good
**Rating:** 8
**Confidence:** 4

**Summary:**

The paper proposed Temporal Hierarchies from Invariant Context Kernels, a method for extracting a hierarchical world model using the proposed Context-specific RSSM (which extends the existing RSSM model (Hafner et al., 2018)) using a coarsely-updating context variable. CRSSM predicts in two levels, selectively updating parts of its latent state (context) sparsely in time. They combine the hierarchical world model (CRSSM) with Dreamer (Hafner et al., 2020) to propose THICK Dreamer, a model-based RL method that combines value estimates from low- and high-level predictions of state and context variables to compute a long-horizon value function. The authors evaluate THICK on MBRL tasks and qualitatively show context switches in their world model.

**Strengths:**

1) The paper is easy to follow and well-structured with the main contributions listed clearly in the introduction. The proposed world model, Context-specific RSSM, makes a simple yet effective addition to RSSM, and the authors contextualize it well within the existing RSSM model. Contrasting with the existing literature in this area, I find the model visualizations to be very clear and self-explanatory.
2) The authors show interesting qualitative results using CRSSM, extensive evaluation of their world model when fitted in Dreamer and PlaNet for MBRL, and provide thorough details about their task setups in the appendix adding to the reproducibility of this work.

**Weaknesses:**

1) Looking at figure 7, there does not seem to be a significant difference between the performance of Dreamer and the proposed method. Can the authors justify the marginal performance improvement?
2) The reported results using Director show close to 0 success on all the tasks. Can the authors explain why was that the case?

**Questions:**

None

**Details Of Ethics Concerns:**

No concerns.

---

> ### Author Response · Authors · 2023-11-19
> **Response to Reviewer yLBk**
>
> We thank the reviewer for their great feedback.
>
> ### Performance improvements
>
> Sparse reward tasks such as MiniHack pose the challenges of exploration and learning long-horizon behavior. With THICK Dreamer we only tackle the latter through long-horizon reward propagation. This amplifies reward signals once rewards are discovered but does not necessarily help discover rewards earlier. This is reflected in Fig. 7. A stronger performance gain is achieved when the high-level guides the low-level by setting subgoals (THICK PLaNet, Fig. 9).
>
>
> ### Director in MiniHack
>
> Thank you for this question. To analyze when Director fails in MiniHack, we carefully checked our implementation and analyzed the performance of Director in more detail. This lead us to find a small inconsistency in our Director implementation affecting the MiniHack tasks. We reran all MiniHack experiments. In the simplest problem (KeyRoomFixed-S5), in which Director sometimes managed to solve the task, Director now consistently solves the task. Thank you for making us revisit Director to catch this.
>
> However, the problem of Director never learning to solve most MiniHack tasks remains and we identified **two problems** that impede Director’s performance in MiniHack, which we now outline and discuss in Suppl. F.3.
>
> One problem of Director is that its learning can be slowed down during early stages of training depending on the **diversity of initial data**. Director trains a goal encoder, which defines the input space for the goal-conditioned policy, on its replay buffer. Thus, if in the beginning not enough diverse data is collected this is reflected in the goal space. As a result, the manager (high-level) does not set meaningful goals for the worker (low-level). We exemplify this for variants of the KeyDoor problem in MiniHack in Suppl. F.3 and Fig. 19. By increasing the diversity of collected data, e.g., randomizing spawn positions and increasing level layout, the performance of Director during initial training strongly improves.
>
> A more severe problem of Director is **unobservable information for setting goals**. The RSSM encodes both observable as well as unobservable information within its deterministic latent state $h_t$. If the unobservable information, e.g., item pick-up in MiniHack, does not directly affect rewards or substantially influence image reconstruction, it might be encoded only locally in $h_t$ and wash out over time. In Dreamer, this is not a problem because the policy can amplify task-relevant information in $h_t$. Director, however, compresses $h_t$ into a low-dimensional goal-encoding. Thereby, task-relevant latent information in $h_t$ could get lost. Note that all novel tasks proposed in the Director paper [Ref a] avoid long-horizon memory, for example by coloring walls in a maze (Egocentric Ant Maze) or by providing a visible history of past button presses (Visual Pin Pad).
>
> In smaller MiniHack tasks, e.g., KeyRoomFixed-S5, this can sometimes be circumvented by specifying goals via observable information. For example, if both the key and door are visible a goal would be the same observation without the key visible (picked up) and an open door. This creates a curriculum, in which the worker can first learn from such simpler situations and later learn to pick up a key and open the door automatically from the high-level goal of an open door. In larger task spaces, e.g. KeyCorridor-8, Director never encounters such simpler situations to begin with.
>
> We thank the reviewer for their review, and we are happy to answer further questions.
>
> ### References
>
> [Ref a] Hafner et al. Deep hierarchical planning from pixels. NeurIPS 2022

---

### Official Review · Reviewer_h1NE · 2023-11-05

**Soundness:** 2 fair
**Presentation:** 4 excellent
**Contribution:** 3 good
**Rating:** 6
**Confidence:** 3

**Summary:**

This paper proposes a novel way of learning a hierarchical world model. On top of Dreamer's world model implementation, RSSM, this paper introduces Context-specific RSSM (C-RSSM) with a slowly (sparsely) changing discrete context state $\mathbf{c}_t$, which represents some static scene info that is preserved for a long time horizon. The continuously changing states $\mathbf{h}_t$ and $\mathbf{z}_t$ are then conditioned on this context state $\mathbf{c}_t$. With the trained C-RSSM, a high-level transition is defined as a contiguous transition segment with the same $\mathbf{c}_t$. The high-level world model (THICK) can be trained to predict the context and stochastic state of the next segment. The paper utilizes C-RSSM and THICK for model-based RL by combining with Dreamer and model-based planning by using MPC. The experiments on MiniHack, PinPad, and robotic manipulation environments show that THICK+Dreamer and THICK+MPC outperform the flat world model baselines when the task horizon becomes longer.

**Strengths:**

* This paper tackles an important problem of learning a hierarchical world model.

* The idea of learning a sparsely changing context state is intuitive and using context switches to define high-level transitions is a sensible choice. Moreover, this enables THICK to naturally incorporate variable-length skills.

* The learned hierarchical world models (THICK and C-RSSM) can be integrated with both model-based RL and model-based planning.

* The paper is clearly written and the figures help us understand the complex concepts of the proposed method.

* The experimental results support that the hierarchy in the world model improves long-horizon prediction.

**Weaknesses:**

* The design choices of the proposed approach are not explained and justified. Especially, the high-level world model learns to predict the future state and action just before the context switch, rather than the context and state right after the context switch. Predicting a low-level action sometime in the future sounds very difficult to me. Although this design choice also makes sense and works in practice, it would be great to explain the rationale behind this specific design choice.

* Section 2.1 introduces the coarse prior of the stochastic state but it is unclear what is the rationale behind this design. It would be great to further explain why C-RSSM needs both coarse and precise priors. Similarly, more explanation about the need for coarse predictions in Equation 8 would help understand the proposed method.

* Generally, the experimental results of THICK+Dreamer and THICK+PlaNet are similar to those of DreamerV2 and PlaNet. Despite its novelty, these weak experimental results may make its impact less significant. Stronger results in more diverse environments would be greatly appreciated. If possible, it would be great to see its performance on the common RL benchmarks, such as Atari and DMC tasks.

* The paper claims that Thick Dreamer is more sample efficient than DreamerV2 in PinPadFour and PinPadFive but the improvement is relatively marginal to strongly claim this.

* Moreover, the experiments on the PinPad environments use Plan2Explore to fill in initial exploratory data. As RL is inherently a combination of exploration and exploitation, it would be important to see how it works for exploration. Thus, it is recommended to include experiments on RL from scratch.

* Although the investigation of world model hierarchies is important, many deep learning approaches seem in favor of scaling model sizes instead of injecting hierarchies. In this sense, it might be interesting to see comparisons between scaling models [a] vs. C-RSSM.


[a] Deng et al. Facing Off World Model Backbones: RNNs, Transformers, and S4. NeurIPS 2023

**Questions:**

Please address the weaknesses mentioned above.


### Minor questions and suggestions

* It might be better if Figure 2 could illustrate that $\mathbf{c}$ is changing slowly and the high-level transition happens when $\mathbf{c}$ has changed.

* In Equation 11, $\delta$ starting from 0 makes more sense?

* In Equation 21, $t < K$ should be $\tau(t) < K$ or $A_{\tau(t):K}$ should be $A_{t+1:K}$?

---

> ### Author Response · Authors · 2023-11-19
> **Response to Reviewer h1NE (1/2)**
>
> We thank the reviewer for their super helpful comments and suggestions and for thoroughly checking our equations and catching some mistakes.
>
>
> ### Design choice 1: Predicting states before context transitions
>
> Our high-level model is a generative model that learns to predict situations in which latent generative factors are assumed to change. Such situations can be related to context transitions [Ref b] or event boundaries [Ref c], as well as to context switches when considering the RL options framework [Ref d]. To predict such situations, two challenges arise: (1.) learning in which situation such transitions are currently possible and (2.) how these transitions affect the latent generative factors. The low-level world model already learns to encode (2). Accordingly, we train the high-level model to only learn (1) and then prompt the low-level model for (2).
>
> We further emphasize the reasons for this design choice now in Sec 2.2 and explain this in detail in a new “design choices” subsection in Suppl. D.1.
>
> ### Design choice 2: Coarse predictions
>
> Thank you for pointing out that we may have not highlighted this sufficiently. The coarse prediction are important for THICK for two main reasons:
>
> **(1.) Prediction-relevant contexts $c_t$:** Context $c_t$ is designed to encode latent information that is necessary for prediction and reconstruction. If we would omit the coarse prior predictions (Eq. 5) and coarse output reconstructions (Eq. 8) the C-RSSM would have no incentive to encode prediction-relevant latent information in $c_t$, reducing the system to the RSSM. As a result, no hierarchy would develop and hierarchical planning would stay impossible.
>
>
> **(2.) Predictions without $h$:**  The high-level model attempts to predict a future state of the system. The full latent state would contain the deterministic component $h$. However, for the high-level model it would be very challenging to predict the unregularized high-dimensional hidden state $h$ many time steps in the future. Thus, it is advantageous that the C-RSSM can make predictions without $h_t$. After a high-level prediction we can simply feed the high-level outputs ($\hat z_{\tau(t) -1}, \hat a_{\tau(t) -1}$) into the low-level world model. This allows us to predict rewards or episode terminations/discounts at particular situations (used in THICK Dreamer and THICK PlaNet) or reconstruct images to visualize predictions (see Sec 3.1).
>
> We emphasize the reasons for this design choice now more clearly in Sec 2.1 and 2.2 and add a paragraph to the Suppl. D.1 to explain these critical design choices in further detail.
>
> ### Common RL benchmarks
>
> THICK world models attempt to tackle partially-observable problems with long task horizons. The problems in the DMC suite have a short task horizon. The problems in Atari are almost (via frame stacking) fully observable. Only a few problems have longer task horizons. Thus, we believe that these suites would not showcase the advantage of using THICK world models particularly well.
>
> ### Exploration data
>
> Please note that in all MiniHack experiments we do not use exploration data. However, with THICK Dreamer we only tackle exploitation through reward propagation and do not use the hierarchy for exploration. This is why we sidestepped the hard challenge of exploration in Visual Pin Pad.
>
> However, we now plot the return curves for PinPadThree without exploration data in Suppl. F.5 and Fig. 20a. For PinPadFour and PinPadFive no rewards are discovered within 600k steps for THICK Dreamer and Dreamer without exploration data.
>
> To paint a full picture,we analyzed the effect of the number of exploration samples on performance by comparing different amounts of exploration samples (250k, 500k, 1000k) in Suppl F.5, Fig. 20 and Tab. 3. The effects of slightly increased sample efficiency for THICK Dreamer in PinPadFour becomes more pronounced when fewer exploration samples are available. Regardless of the amount of exploration data THICK Dreamer always matches or outperforms its flat counterpart.
>
> ### Comparison model size vs. hierarchy
>
> Thank you for the excellent suggestion. Unfortunately, we could not compare against [Ref a] since their code is not publicly available. But in a new additional experiment (Suppl. F.7 and Fig. 22), we compare Dreamer and PlaNet with different RSSM model sizes against THICK Dreamer and THICK PlaNet in MiniHack-WandOfDeathAdv and Multiworld-Door. We change model size by modifying the number of hidden units per layer and the size of the deterministic hidden state $h_t$ by different factors ($\in \{0.5, 1, 2, 4\}$ for S-, M-, L-, XL-sized models). Increasing model size does not improve Dreamer or PlaNet in these tasks. Our unmodified THICK world models always achieve the best performance.

---

> > ### Author Response · Authors · 2023-11-19
> > **Response to Reviewer h1NE (2/2)**
> >
> > ### Suggestions and corrections
> >
> >
> > > The paper claims that Thick Dreamer is more sample efficient than DreamerV2 in PinPadFour and PinPadFive but the improvement is relatively marginal to strongly claim this.
> >
> > We reduced the claim accordingly.
> >
> > > Figure 2 could illustrate that $c$ is changing slowly and the high-level transition happens when $c$ has changed
> >
> > Fig. 2 shows the C-RSSM, which can be used independently of THICK and the high level. Thus, we refrain from adding the high-level training process to this figure. Instead, we hope that Fig. 3 illustrates inputs/targets of the high-level with Fig. 3a illustrating the sparse context changes.
> >
> > > In Equation 11, starting from $\delta=0$ makes more sense
> >
> > Yes, this is correct. Thank you for catching this typo.
> >
> > > In Equation 21, should be [...] $A_{t+1:K}$?
> >
> > Thank you for pointing this out. In our original formulation it should have been $A_{t+1:K}$. However, we realized that in Eq. 21 using $t$ as a subscript for high-level actions $A$ may be misleading because the planning depth of the high-level planning does not depend on the actual time scale. Thus, we modified Eq. 21 and replaced $A_{t:k}$ with $A_{k:K}$.
> >
> > Thank you very much for thoroughly checking our equations.
> >
> >
> > ### References:
> >
> > [Ref a] Deng et al. Facing Off World Model Backbones: RNNs, Transformers, and S4. NeurIPS 2023
> >
> > [Ref b] Heal et al., Contextual inference underlies the learning of sensorimotor repertoires. Nature, 2021
> >
> > [Ref c] Radvansky & Zacks. Event cognition. Oxford University Press, 2014
> >
> > [Ref d] Sutton et al., Between MDPs and semi-MDPs: A framework for temporal abstraction in reinforcement learning. Artificial Intelligence, 1999.

---

> > > ### Comment · Reviewer_h1NE · 2023-11-22
> > > **Thank for author responses**
> > >
> > > Thank the authors for the detailed responses and updates in the paper!
> > >
> > > **Design choice 1: Predicting states before context transitions**
> > >
> > > I understand that reusing the low-level world model could be efficient. However, this is better only when predicting $\mathbf{z}_{\tau(t)-1}, \mathbf{a}_{\tau(t)-1}$ is easier than directly predicting $\mathbf{c}_{\tau(t)}$. I don't quite get why the former is better and the latter is worse. It would be great if the authors could elaborate on this in the paper.
> > >
> > > **Common RL benchmarks**
> > >
> > > My suggestion for the common RL benchmarks is not to show the benefit of THICK dreamer on these tasks but to show its generality (i.e. THICK dreamer can work on diverse environments), hopefully not loosing too much performance on them. This can provide a good indicator for readers when to expect performance gains and when not to use this approach, if any.

---

> > > > ### Author Response · Authors · 2023-11-22
> > > > **On the design choice of predicting states before context transitions**
> > > >
> > > > Thank you for the follow-up question and allowing us to clarify this better.
> > > >
> > > > We use the high-level prediction for anticipating rewards, discounts, or reconstructing images in the future $\tau(t)$. Note that currently the context $c_{\tau(t)}$ alone is too coarse for these predictions and we additionally need the stochastic state $z_{\tau(t)}$ (see Eq. 8). Thus, currently we anyway need to predict $z$ on the high level. We argue that it is easier to predict the observable information right before the transition, i.e., $z_{\tau(t)-1}, a_{\tau(t)-1}$, and let the C-RSSM take care of the context transition, than predicting stochastic state and context after the context transition. For example, in MiniHack our system would need to predict the agent standing in front of closed door and performing the OPEN-action as opposed to predicting the view of the agent after opening a door and looking into a (potentially unknown) room.
> > > >
> > > > However, we agree that this is a design choice worth discussing. Predicting information after the transition is also an interesting direction. For example, in future work external quantities, such as rewards or discounts, could be directly predicted from $c_t$. We updated the paper again to further motivate and discuss this design choice in Sec. 2.1-2.2, in Suppl. D.1. and Sec. 5. Additionally, we illustrate that our high-level model can predict meaningful low-level actions before context transitions by showing exemplary actions predicted by the high level for MiniHack-EscapeRoom in Fig. 14 and describing them in Suppl. F.1. We hope this further illustrates that our hierarchical design is sensible.

---

### Official Review · Reviewer_sX3c · 2023-11-09

**Soundness:** 3 good
**Presentation:** 3 good
**Contribution:** 3 good
**Rating:** 6
**Confidence:** 3

**Summary:**

This paper proposes a method called THICK for learning hierarchical world models with adaptive temporal abstractions from discrete latent dynamics. The key idea is to use a context-specific recurrent state space model (C-RSSM) to create a sparsely changing context latent variable. This context encodes higher-level transitions that are used to train a high-level predictor. The high-level model predicts states that lead to context changes, enabling temporal abstract predictions. The hierarchical predictions can be integrated into model-based RL and planning methods.

**Strengths:**

- The C-RSSM provides a simple yet effective way to learn sparsely changing context variables from pixel observations without supervision.

- The high-level model is trained in a self-supervised manner to anticipate context changes based on the C-RSSM's discrete dynamics.

- The method is generally applicable across various environments with visual observations.

- Experiments show that the learned hierarchies capture meaningful abstractions related to subgoals.

- Integrating THICK's hierarchical predictions improves sample efficiency of model-based RL in long-horizon tasks.

- The high-level plans can be visualized, enhancing model interpretability.

**Weaknesses:**

1) The sparsity hyperparameter for the C-RSSM needs to be tuned for each environment.

2) The high-level model operates at a fixed abstract timescale, less flexible than methods with hierarchical policies.

3) The high-level plans are not actively updated during execution, being replanned only at context changes.

4) The expressiveness of temporal abstractions may be limited compared to methods with backing task priors or curriculum learning.

However, I think this paper still take a step further to hierarchical world model.

**Questions:**

1) How does the performance compare to hierarchical RL methods like h-DQN or FeUdal Networks?

2) Could the hierarchy be extended to have multiple abstraction levels instead of just two?

3)  How well does THICK scale to even longer time horizons or higher-dimensional state spaces?

4)  Could active exploration be used to shape useful context abstractions instead of relying on a sparsity loss?

5) Is there some mechanisms could be add to make the high-level plans more reactive to execution errors or environment changes?

---

> ### Author Response · Authors · 2023-11-19
> **Response to Reviewer sX3c**
>
> Thank you for the valuable feedback and questions. First, we would like to clarify potential misunderstandings before answering questions.
>
>
> ### Sparsity loss scale
> While it is true that ideally $\beta^\mathrm{sparse}$ can be set individually for each task, in practice the same value works well across similar tasks (e.g., same setting per suite, see Suppl B).
>
>
> ### Time scale of the high-level
>
> It is not true that the high level operates on a fixed time scale. This is one important feature of THICK - we apologize for not clarifying this sufficiently well.  As we explain in Sec 2.2, “THICK uses [...] discrete context dynamics as an **adaptive** timescale for training a high-level world model”. The function $\tau(t)$ implements a **variable** temporal abstraction that maps a point in time to the next point in time with context changes. Thus, the high-level learns **predictions of variable length**, anticipating context transitions. As we analyze in Suppl. F.1 (e.g., Figure 16) the duration between context changes can widely vary within a task or across tasks. We highlight this now more in the text.
>
> ### Reactivity of high-level plans
> In the paper, we had THICK PlaNet replan on the higher level only when the lower level context changes. This decreases computational effort. In a new ablation (Suppl F.8 & Fig. 27) we show that replanning on the higher level in every step does not drastically change performance in Multiworld-Door. Thus, while THICK PlaNet may run model-based planning on the higher level in every time-step, we recommend running it sparsely or only when context changes to save computational energy.
>
> ### Task priors or curriculum learning
> We unfortunately do not understand this point. Could you please elaborate? We do not see curriculum learning and our learning of temporal abstractions to be exclusive. They could be easily combined.
>
> ### Questions
>
> > How does the performance compare to hierarchical RL methods like h-DQN or FeUdal Networks?
>
> h-DQN requires pre-determined abstractions [Ref a], white our method discovers them from scratch.
>
> FUN performs worse or less sample efficiently than DreamerV2 when both methods are applied to the same problems (compare Fig. 4 of [Ref b] with Figure F.1 in [Ref c], or compare Fig. 2 of [Ref b] with Fig. B.2 in [Ref c]). Seeing that THICK Dreamer performs as good or better than DreamerV2, we believe it would outperform FUN.
>
> > Could the hierarchy be extended to have multiple abstraction levels instead of just two?
>
> As we write in our conclusion, THICK is not restricted to learning a two-level hierarchy. By replacing level 2 with a C-RSSM, in principle, the same mechanism could be applied to learn a level 3 world model etc. In this way an N-level hierarchy of world models with nested time scales could develop.
>
> Another option would be to foster the development of  multiple coarse pathways on the low level by means of varying sparsity regularization strengths. For each pathway a separate high-level model could be trained. In this approach, the developing hierarchy would not necessarily be strictly nested.
>
> > How well does THICK scale to even longer time horizons or higher-dimensional state spaces?
>
> We believe THICK should scale to very long time horizons, since it does not depend on the exact time scale of the environment but on the developing time scale of context updates. Similarly, we expect THICK to be also applicable to problems with high-dimensional state spaces. Potentially the dimensionality of $c_t$ would need to be increased to capture all possible context changes in the environment.
>
> > Could active exploration be used to shape useful context abstractions instead of relying on a sparsity loss?
>
> This is an excellent point. Active exploration might support the shaping of context abstractions. We keep this as future work, also as it would be a complementary direction.
>
> > Is there some mechanisms could be add to make the high-level plans more reactive to execution errors or environment changes?
>
> As outlined above and shown in our new ablation in Suppl. F.8 and Fig. 27, high-level planning could be active at every time step
>
>
> ### References:
>
> [Ref a] Kulkarni et al. Hierarchical Deep Reinforcement Learning: Integrating Temporal Abstraction and Intrinsic Motivation. NeurIPS 2016
>
> [Ref b] Vezhnevets et al. FeUdal Networks for Hierarchical Reinforcement Learning. ICML 2017
>
> [Ref c] Hafner et al. Mastering Atari with Discrete World Models. ICLR 2020

---

### Author Response · Authors · 2023-11-19
**General answer**

We thank the reviewers for their overall very positive comments, the valuable feedback and suggestions on our novel THICK world models.

### Our reply in short (details below):

We have revised our paper trying to include all the reviewer’s concrete questions and optimizing the flow and readability on the way.

In our revision we have added the following:
- A deeper analysis of the Director model, including insights on why it fails on some of the considered problems;
- Another ablation study showing THICK PlaNet’s robustness against the frequency of high-level planning;
- An evaluation varying exploration strength during training, confirming and underlining THICK Dreamer’s better performance;
- Comparisons with different RSSM sizes, yet again confirming THICK’s better performance.

We need to also emphasis a correction and an improved study concerning the Director system:

### Director performance in MiniHack (yLBk):

By analyzing why Director performs worse in MiniHack we discovered a small inconsistency in our Director implementation affecting MiniHack. We apologize for this error. We ran all Director experiments in MiniHack again. In one task (KeyRoomFixed-S5), which Director previously sometimes (but not consistently) learned to solve, Director now achieves strong performance. In the remaining tasks, however, Director’s performance remained nearly unchanged. Thus, the main message of this experiment concerning Director’s performance, i.e., that Director fails to learn most MiniHack tasks, stays the same. To further analyze why Director fails in the other task, we added an additional analysis in Suppl. F.3 and conclude that (i.) Director strongly benefits from sources of randomness during training and (ii.) seems to fail to specify unobservable information within its goals.

### Further experiments and descriptions (sX3c & h1NE):

Further additions in our revised version include:

-  We provide a new **ablation of THICK PlaNet updating high-level plans in every time step** in Suppl. F.8 & Fig. 27 (sX3c). The results show that THICK PlaNet does not show worse performance when the high level sets a new goal in every time step.
- We have **improved the description of design choices** in the main text (Sec. 2.2 & 2.3) and in the dedicated supplementary section D.1 (sX3c & h1NE)
- We compare performance in **Visual Pin Pad without exploration or fewer exploration steps** in Supp F.5 and Fig. 20 (h1NE). The results show that both THICK Dreamer and Dreamer require sufficient exploration mechanisms for larger problems. Meanwhile, THICK Dreamer always matches or outperforms its flat counterpart.
- We also now **compare THICK’s hierarchical approach to flat approaches with increased model size** in Suppl. F.7 and Fig. 22 (h1NE). The results show that larger model sizes do not improve Dreamer and PlaNet to match the THICK versions.


We hope the more detailed responses to the individual reviews address all the remaining questions raised. Thank you once again so much for your time, effort, and consideration!

---

### Meta-Review · Area_Chair_jtrz · 2023-12-14

**Metareview:**

This paper presents a novel hierarchical MBRL algorithm using discrete representations between the levels of the hierarchy.

The topic of this contribution is timely and relevant to the community.

Overall, the manuscript is a good contribution to the literature, and the experimental results are thorough.

I encourage the authors to incorporate the feedback from the reviewers in the final manuscript.

**Justification For Why Not Higher Score:**

While technically strong, it is not obvious that this paper will have a large impact on the RL community.

**Justification For Why Not Lower Score:**

The contribution of this paper is timely and interesting. The experimental results are thorough.

---

### Decision · Program_Chairs · 2024-01-16

Accept (spotlight)